# Social isolation shortens lifespan through oxidative stress in ants

Akiko Koto [1,2] ✉, Makoto Tamura [3], Pui Shan Wong [2], Sachiyo Aburatani [1,2], Eyal Privman [4], Céline Stoffel [5], Alessandro Crespi [6], Sean Keane McKenzie [5], Christine La Mendola [5], Tomas Kay [5] & Laurent Keller [5,7] ✉

Social isolation negatively affects health, induces detrimental behaviors, and shortens lifespan in social species. Little is known about the mechanisms underpinning these effects because model species are typically short-lived and non-social. Using colonies of the carpenter ant *Camponotus fellah*, we show that social isolation induces hyperactivity, alters space-use, and reduces lifespan via changes in the expression of genes with key roles in oxidation-reduction and an associated accumulation of reactive oxygen species. These physiological effects are localized to the fat body and oenocytes, which perform liver-like functions in insects. We use pharmacological manipulations to demonstrate that the oxidation-reduction pathway causally underpins the detrimental effects of social isolation on behavior and lifespan. These findings have important implications for our understanding of how social isolation affects behavior and lifespan in general.

Social environments structure the lives of social organisms, and social isolation can be detrimental to health[1–4]. Social isolation can decrease lifespan[5,6], reduce immune capacity[7,8], disrupt sleep[9–12], and cause metabolic dysfunction[13]. Recent studies have suggested that neural[14–17] and epigenetic changes[18–20] are involved in the detrimental effects of social isolation, but the mechanisms remain poorly understood[3], because most of the model species that can readily and ethically be kept in social isolation are non-social.

Eusocial insects (mostly ants, some bees, some wasps, and termites) naturally live in complex societies and so represent useful models to study how social environments influence longevity. The evolution of social life was accompanied by a 100-fold increase in lifespan of reproductive queens relative to their solitary ancestors[21]. Queens can live over 20 years in some species[22,23], about 10 times longer than non-reproductive workers[21,24]. Several social factors are known to affect lifespan. For example, the removal of queens leads to ovarian development and longer lifespans of worker ants[25–27] and

honeybees[28], suggesting that longevity is tightly linked with reproductive capacity and social environment. Social isolation has also been shown to decrease lifespan in several eusocial lineages, including ants, termites, and bees[29]. For example, isolated workers of the carpenter ant *Camponotus fellah* have a shorter lifespan than workers kept in groups with no queen or brood[30,31]. In *C. fellah*, isolation also leads to important behavioral changes (e.g., increased motor activity and a tendency to leave the nest and stay in peripheral zones of the foraging arena) and an imbalance in energy intake and expenditure[30,31]. Social isolation also shortens lifespan and impairs behavior and physiology of socially isolated individuals of other ant species (e.g., *Temnothorax nylanderi*[32] and *Solenopsis invicta*[33]). The effect of social isolation is directly mediated by changes in the digestive processes[31]. For example, labeled food is retained longer in *Formica* ant workers when they are grouped rather than isolated, possibly due to food exchange among workers[34]. Digestion is also influenced by factors such as temperature, starvation period, and food type[35]. The effect of social isolation is also

[1]Bioproduction Research Institute, National Institute of Advanced Industrial Science and Technology, Tsukuba 305-8566 Ibaraki, Japan. [2]Computational Bio Big Data Open Innovation Laboratory (CBBD-OIL), National Institute of Advanced Industrial Science and Technology, Tsukuba 305-8566 Ibaraki, Japan. [3]NeuroDiscovery Lab, Mitsubishi Tanabe Pharma America, Cambridge, MA 02139, USA. [4]University of Haifa, Institute of Evolution, Department of Evolutionary and Environmental Biology, Haifa 3498838, Israel. [5]University of Lausanne, Department of Ecology and Evolution, Lausanne CH-1015, Switzerland. [6]Biorobotics Laboratory, Ecole Polytechnique Fédérale de Lausanne, Lausanne CH-1015, Switzerland. [7]Present address: Social Evolution Unit, Cornuit 8, BP 855, Chesières CH-1885, Switzerland. ✉e-mail: a-koto@aist.go.jp; Laurent.keller01@gmail.com

indirectly mediated by social interactions. The presence of larvae has a positive effect on lifespan of several species[31,32].

Several studies have also started to investigate the mechanisms that mediate the reduced longevity of socially-isolated individuals. For example, it has been shown that social isolation affects the expression of genes related to immune function, stress response[36,37] and levels of biogenic amines[38–40]. However, the mechanisms remain poorly understood because until recently it has been difficult to conduct genetic and pharmacological manipulations in social insects.

In this study, we use *C. fellah* as a model species to investigate the molecular and physiological mechanisms underlying the detrimental effects of social isolation. We performed behavioral tracking[41] and RNA-sequencing (RNA-seq) to study the transcriptomic differences between workers kept in social isolation versus workers kept in groups. We also examined how differences in behavior between isolated ants correlated with differences in gene expression. We found that the social isolation-induced changes in physiology and behavior are primarily mediated by changes in the expression of key oxidoreductase genes and the accumulation of reactive oxygen species (ROS). We showed that higher ROS production was most pronounced in the peripheral fat body and hepatocyte-like cells called the oenocytes. We therefore tested whether ROS detoxification could extend the lifespan of socially isolated ants. We found that the administration of antioxidant compounds such as melatonin, prevented the accumulation of ROS and rescued the ants from the reduction in lifespan and detrimental behaviors normally associated with social isolation.

## Results

### Identification of isolation-related differentially expressed genes and enriched pathways

We used an automated system of behavioral tracking, gluing unique ARTag barcodes to the thorax of each individual and using monochrome digital video cameras (2 frames/s)[41–43] to characterize the behavior of workers kept for 24 h in groups of ten (hereafter refered to as grouped) or in social isolation (refered to as isolated). In this experiment, we aimed to identify genes whose expression varied with social condition by performing behavioral analysis and RNA-seq at the individual level. We used 4-month-old workers because older workers contain high levels of formic acid, which decreases the amount of purified RNA that can be extracted. We quantified the time spent by workers in five regions (nest, arena, near the walls, food area, and water area; Fig. 1a, Supplementary Fig. 1a, b). Consistent with previous results[31], social isolation led to a shift in space use, increasing the amount of time spent near the walls ($F_{20} = 29.5$, $p < 0.0001$), in the arena ($F_{21} = 15.5$, $p < 0.001$) and in the water area ($F_{20} = 5.5$, $p = 0.029$), and decreasing the time spent in the nest ($F_{21} = 36.4$, $p < 0.0001$). Changes in activity levels of isolated individuals have also been reported in other social and non-social insects[4,44–46]. Because ants spent most of their time in the nest or near the wall, we later use the ratio of time between these locations as a behavioral indicator of social isolation (Figs. 1g, 2a, c, e, g, 3c, and 4d). There was a strong negative correlation between the time spent near the wall and time spent in the nest ($F_{49} = 358.5$, $p < 0.0001$, Fig. 1b). In addition to changing space use patterns, social isolation led to hyperactivity, increasing speed ($F_{21} = 7.87$, $p = 0.011$, Fig. 1c) and total distance covered during the experiment ($F_{20} = 21.6$, $p < 0.001$, Fig. 1c).

After 24 h of automated tracking, RNA was extracted from the whole body of each ant to perform RNA-seq. Social isolation led to the downregulation of 487 and the upregulation of 407 genes (Benjamini–Hochberg false discovery rate (FDR) corrected *p*-values (i.e., *q*-values) <0.05, Fig. 1d; Fig. 1e for the 60 differentially expressed genes (DEGs) with absolute log₂ fold changes ≥1, Supplementary Fig. 1c for all DEGs). Gene Ontology (GO) analysis revealed that these genes were most significantly enriched for oxidoreductase activity (*q*-value = $4.72 \times e^{-09}$, Fig. 1f). To further investigate how social isolation

affects gene expression, we performed a weighted gene co-expression network analysis (WGCNA)[47], which describes correlational patterns between genes and provides biological interpretations of gene modules. The resulting network consisted of 16 gene co-expression modules, among which M1 and M11 were positively correlated with the relative amount of time isolated workers spent near the wall, and M9 and M15 were positively correlated with the relative amount of time grouped workers spent near the wall (Fig. 1g). GO analysis of the DEGs in M1 revealed an overrepresentation of three molecular functions, with oxidoreductase activity having the highest bias (*q*-value = $8.86 \times e^{-06}$, Fig. 1h). GO analysis of the DEGs in M11 revealed no significant molecular function, likely because this module contains few DEGs ($n = 34$).

### Differential expression of genes implicated in oxidoreductase activity under social isolation

Since oxidoreductase activity was the most overrepresented GO term (Fig. 1f, h) and because oxidative stress is involved in aging, disease[48] and various behavioral stress responses such as sleep loss[12,49,50], we focused on enzymes with oxidoreductase activity to determine whether they could drive the differences in lifespan and behavior between grouped and isolated workers. In these and later experiments we used >7-month-old workers because they have a lower life expectancy than 4-month-old workers[31], allowing greater replication. Importantly, however, we previously showed that social isolation has similar negative effects on lifespan regardless of worker age[31]. We first determined the expression pattern of the four genes with oxidoreductase activity in M1 with the highest *q*-values and absolute log₂ fold change >1 (Supplementary Data 1). One of these genes (*DUOX*) is a ROS-producing NADPH oxidase[51] (Supplementary Fig. 2, Supplementary Code 2 and 3). Consistent with the view that isolated ants suffer higher levels of stress and higher rates of ROS production, *DUOX* was significantly over-expressed in isolated ants (Fig. 2a). Moreover, there was also a strong positive correlation between level of *DUOX* expression and relative time spent near the wall (Fig. 2a).

The three other genes are implicated in detoxification. *Wwox-like 1* belongs to a large family of oxidoreductases (Supplementary Figs. 3a, b, Supplementary Code 4 and 5). A homolog in mice (NADP-retinol dehydrogenase) is known to perform oxidation-reduction in the retinoid (visual) cycle[52], and to detoxify toxic lipid peroxidation products such as 4-Hydroxynonenal (4-HNE), an electrophilic aldehyde used as the biomarker of oxidative stress[53]. The two remaining genes (*CYP336A26* and *CYP6AQ19*) belong to the cytochrome P450 (CYP) superfamily, which carries out a diverse range of enzymatic reactions[54]. *CYP336A26* belongs to the CYP336 subfamily, which is related to the CYP28 subfamily, and *CYP6AQ19* belongs to the CYP6 subfamily (Supplementary Figs. 4a, b, and Supplementary Code 6–9). These CYP subfamilies include enzymes that detoxify various chemicals, including natural metabolites and pesticides[55–57]. In line with the notion that isolated individuals, particularly those that spend a higher proportion of time near the wall, suffer from high levels of oxidative stress, the level of expression of these three genes was significantly lower in isolated ants (Fig. 2c, e, g) and was strongly negatively correlated with the relative amount of time spent near the wall.

To investigate the patterns of expression of these genes, we performed quantitative RT-PCR (qRTPCR) on the heads and abdomens of an additional set of grouped and isolated >7-month-old workers. For all four genes, social isolation affected patterns of gene expression in the the same direction as in the RNA-seq experiment, but the differences were not always significant and the magnitude of the effect varied by tissue. For two genes (*CYP336A26* and *CYP6AQ19*), the differences in expression between grouped and isolated ants were significant only in the abdomen (Fig. 2d, h) while for the two remaining genes (*DUOX* and *Wwox-like 1*) the difference was significant in both body parts (Fig. 2b, f). Next, we asked whether the differential expression of these genes was localized to specific tissues within the abdomen. We focused on

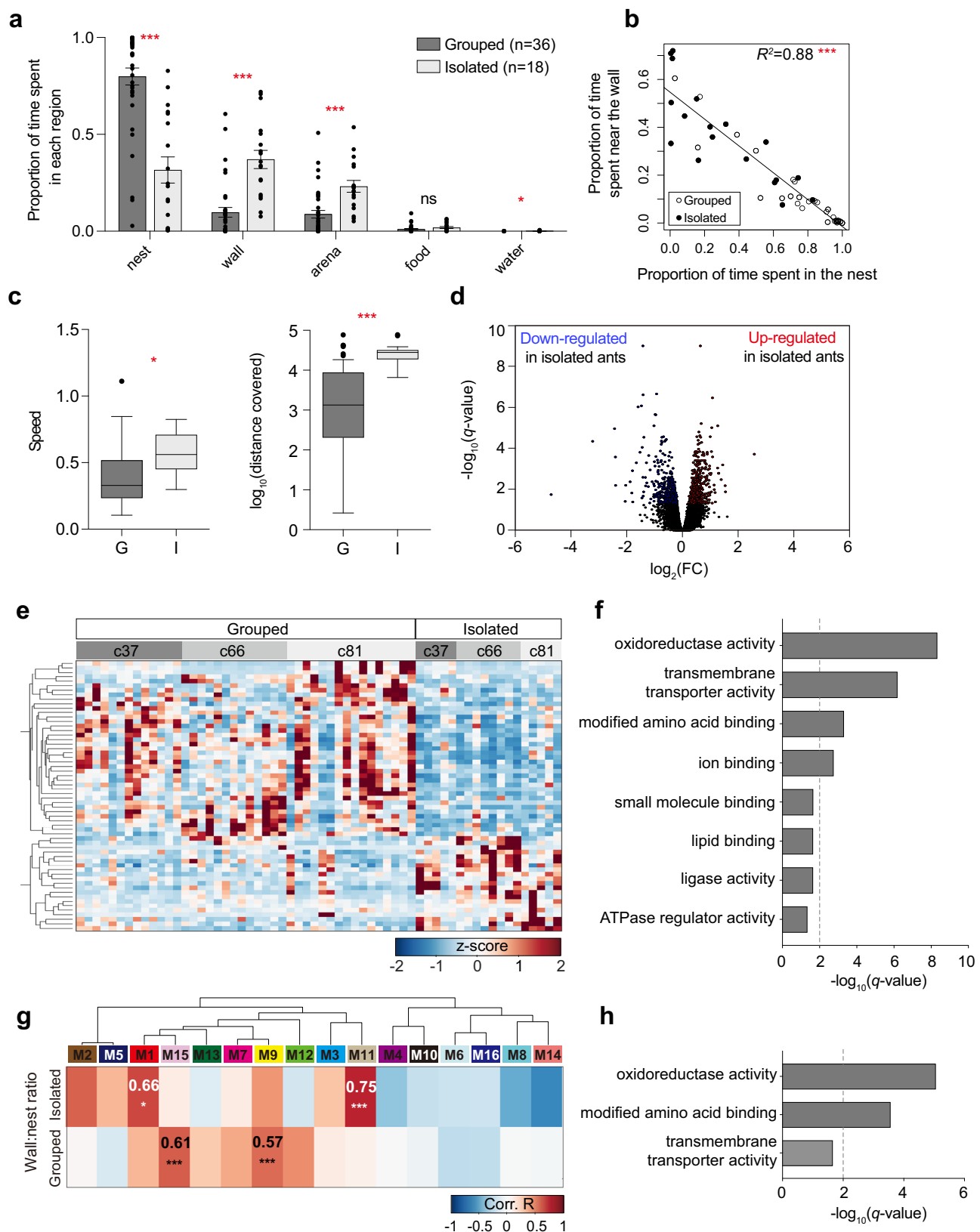

the digestive tract, the fat body, and hepatocyte-like cells called oenocytes[58,59], which are interspersed within the fat body in ants[60]. Because it is technically difficult to separate oenocytes from the fat body, we extracted RNA from the fat body and oenocytes together. Socially isolated ants had significantly lower expression of *Wwox-like 1* and *CYP6AQ19* (Fig. 2f, h) and higher expression of *DUOX* (Fig. 2b) in

the fat body and oenocytes. In the digestive tract, the expression difference was significant only for *DUOX* (Fig. 2b, d, f, h; *CYP6AQ19* was too lowly expressed to be detected by qRTPCR). Thus, social isolation has a greater effect on the expression of genes implicated in the production and detoxification of ROS in the fat body and oenocytes than in the digestive tract.

**Fig. 1 | Differential gene expression analysis and co-expression network under social isolation. a** Behavior profiles of grouped (dark gray, $n = 36$) and isolated (light gray, $n = 18$) ants. The proportion of time spent in five regions (nest: $p < 0.0001$, wall: $p < 0.0001$, arena: $p < 0.001$, food: $p = 0.072$, water: $p = 0.029$) was estimated from 24 h of automated tracking (mean ± SEM with all data points). **b** Correlation between the time spent near the wall and time spent in the nest ($p < 0.0001$) in grouped (open circle, $n = 36$) and isolated (filled circle, $n = 18$) ants. $R$-squared value from a simple linear model. **c** Box plots of speed ($p = 0.011$, cm/frame) and distance covered ($p < 0.001$, cm/day) by grouped (G, dark gray, $n = 36$) and isolated (I, light gray, $n = 18$) ants during 24 h of automated tracking. Boxes and median lines represent inter-quartile range and median values, and whiskers represent minimum and maximum values of data within 1.5-fold of the inter-quartile range. Points indicate outliers. The effect of treatment on behavior in (**a**–**c**) was tested using generalized linear mixed models (GLMMs) and subsequent ANOVA tests: *$p < 0.05$; ***$p < 0.001$; ns, not significant. **d** Volcano plot of expression fold change and FDR corrected $p$-values ($q$-values) for DEGs between grouped and isolated ants. The blue dots denote genes that are downregulated in isolated ants, and the red dots denote genes that are upregulated in isolated ants. **e** Heatmap with hierarchical clustering of 60 genes differentially expressed by social treatment ($q$-values $< 0.05$; absolute $\log_2$ fold change $\geq 1.0$). Each row corresponds to one gene and each column to one sample. The values in the heatmap are the gene expression z-scores. Colony ID is indicated at the top of each column. **f** Molecular function GO terms enriched among the 894 genes affected by social treatment. The dashed line indicates the threshold of $q = 0.01$. **g** Correlations between module eigengenes (MEs), which represent the first principal component of the gene expression in each module computed with WGCNA and wall:nest ratio in isolated ants (top), or in grouped ants (bottom). The values in the heatmap are Pearson's correlation coefficients. Correlations are significant for two modules of isolated ants (M1; $p = 0.048$, M11; $p = 0.0056$), and two modules of grouped ants (M9; $p = 0.005$, M15; $p = 0.0015$). Stars indicate significant correlations: *$p < 0.05$; ***$p < 0.001$, Bonferroni-corrected two-sided Pearson correlation tests. **h** The GO terms enriched in DEGs in module 1 from the ontology of molecular functions. The dashed line indicates the threshold of $q = 0.01$.

The most important difference between the RNA-seq and qRTPCR data was for *CYP336A26*, which showed a clear difference in expression level between whole bodies of grouped and isolated ants in the RNA-seq data (Fig. 2c) but a significant difference only in the abdomen in the qRTPCR data (Fig. 2d). This difference may stem from tissue-specific differences (e.g., the thorax was not analyzed in the qRTPCR analyses) or, more likely, an age effect (4-month-old workers were used for the RNA-seq analyses and >7-month-old workers were used for the qRTPCR analyses, Fig. 2c, d). Importantly, however, our analyses confirmed that the direction of differential gene expression of the three other genes (*DUOX*, *Wwox-like 1*, and *CYP6AQ19*) was similar for the RNA-seq and qRTPCR analyses, regardless of worker age (Fig. 2a, b, e, f, g, h).

## ROS accumulation and cellular damage in the fat body and oenocytes

The detected effects of social isolation on gene expression should lead to higher ROS levels in isolated ants. To test this, we quantified levels of hydrogen peroxide, which is one of the major active oxygen species causing oxidative stress[61]. While there was no significant difference between the heads and digestive tracts of grouped and isolated ants, social isolation led to significantly higher ROS levels in the fat bodies and oenocytes of isolated workers ($p < 0.001$, Fig. 3a). Histological staining of ROS with CellROX, a reagent which fluoresces upon oxidation by reactive oxygen species also revealed higher ROS signals in both the oenocytes (recognized by the round-shaped nuclei, and indicated with arrowheads in Fig. 3b) and trophocytes which are the main cell-type in the fat body (recognized by their irregular nuclei, and indicated with arrows in Fig. 3b) of isolated ants (top in Fig. 3b, $p < 0.0001$ in oenocytes). Because the fat body and oenocytes are involved in lipid storage and metabolism[58], we next examined the expression of 4-hydroxynonenal (4-HNE), which increases in abundance during oxidative stress via the lipid peroxidation chain reaction[61]. The level of 4-HNE (labeled with anti-4-HNE antibody) was also higher in the oenocytes and trophocytes of isolated workers (middle in Fig. 3b, $p < 0.0001$ in oenocytes). Finally, consistent with the spatial pattern of ROS accumulation in the oenocytes and trophocytes, the expression of the necrosis marker SYTOX Green was higher in both the oenocytes and trophocytes of isolated versus grouped ants (bottom in Fig. 3b, $p < 0.0001$ in oenocytes). The expression levels of the three markers of oxidative stress were consistently higher in oenocytes than trophocytes. Overall, these results indicate that social isolation increases ROS accumulation and cellular damage in the fat body and oenocytes.

To investigate whether the higher level of oxidative stress of isolated ants could result from their hyperactivity, we performed CellROX quantification in the fat body with oenocytes of grouped and isolated individuals after 24 h of behavioral tracking (Fig. 3c). ROS levels were significantly correlated with the ratio of time spent near the wall to time spent in the nest for isolated ants ($R^2 = 0.14$, $F_{33} = 5.88$, $p = 0.021$), but not grouped ants ($R^2 = 0.006$, $F_{35} = 0.52$, $p = 0.48$). However, there was no significant correlation between ROS levels and either speed or distance covered for grouped ants (speed: $R^2 = 0.05$, $F_{35} = 3.1$, $p = 0.085$, distance: $R^2 = 0.003$, $F_{35} = 0.2$, $p = 0.66$) or isolated ants (speed: $R^2 = 0.02$, $F_{32} = 0.5$, $p = 0.47$, distance: $R^2 = 0.015$, $F_{33} = 0.43$, $p = 0.51$) (Fig. 3c). This suggests that greater ROS production is not a result of increased activity. These data also suggest that the isolated workers which spent the most time near the wall (i.e., which exhibited the strongest behavioral response to social isolation) are also the workers that exhibited the greatest ROS production, although the relatively low correlation coefficients demonstrate that other intrinsic factors must also be involved.

## ROS accumulation under social isolation reduces lifespan and leads to abnormal behavior

To assess whether changes in oxidoreductase activity and ROS accumulation causally underpin the reduction in lifespan associated with social isolation, we tested whether ROS detoxification could extend the lifespan of socially isolated ants. To do so, we added one of two antioxidants to the water supply of experimental but not control ants. We tested melatonin, which acts as a radical scavenger and/or antioxidant in both flies[50,62,63] and honeybees[64], and nicotinamide adenine dinucleotide (NAD), which has antioxidant properties in vitro[65] and in vivo[50]. Supplementation of the two antioxidants increased lifespan at one or both of the tested concentrations, with the differences being significant for melatonin at 1 µg/ml ($p = 0.0024$, Fig. 4a) and 0.3 µg/ml ($p = 0.042$, Fig. 4a), and for NAD at 0.13 µg/ml ($p = 0.026$, Fig. 4a). Importantly, treatment with melatonin and NAD had no significant effect on the lifespan of grouped ants (Fig. 4a). If anything, there was a tendency for treatments with the highest dosage of melatonin (1 µg/ml) and both dosages of NAD (0.13 and 0.43 µg/ml) to reduce the lifespan of grouped ants. These data show that antioxidants only have a beneficial effect on lifespan for socially isolated ants in our experiments.

Next, we tested whether 1 µg/ml melatonin, the treatment that most extended the lifespan of isolated ants, could also reduce ROS accumulation and behavioral abnormalities of socially isolated ants (Fig. 4b). Melatonin treatment had a small and non-significant effect on ROS levels in the head and digestive tract but cleared ROS from the fat body and oenocytes in isolated ants ($p = 0.0051$, Fig. 4c). Melatonin treatment also had a significant effect on the ratio of time spent near the wall to time spent in the nest ($F_{64} = 8.5$, $p = 0.005$), but no significant effect on either speed ($F_{50} = 0.005$, $p = 0.94$) or distance covered ($F_{33} = 0.17$, $p = 0.68$) (Fig. 4d). Importantly, these data show that

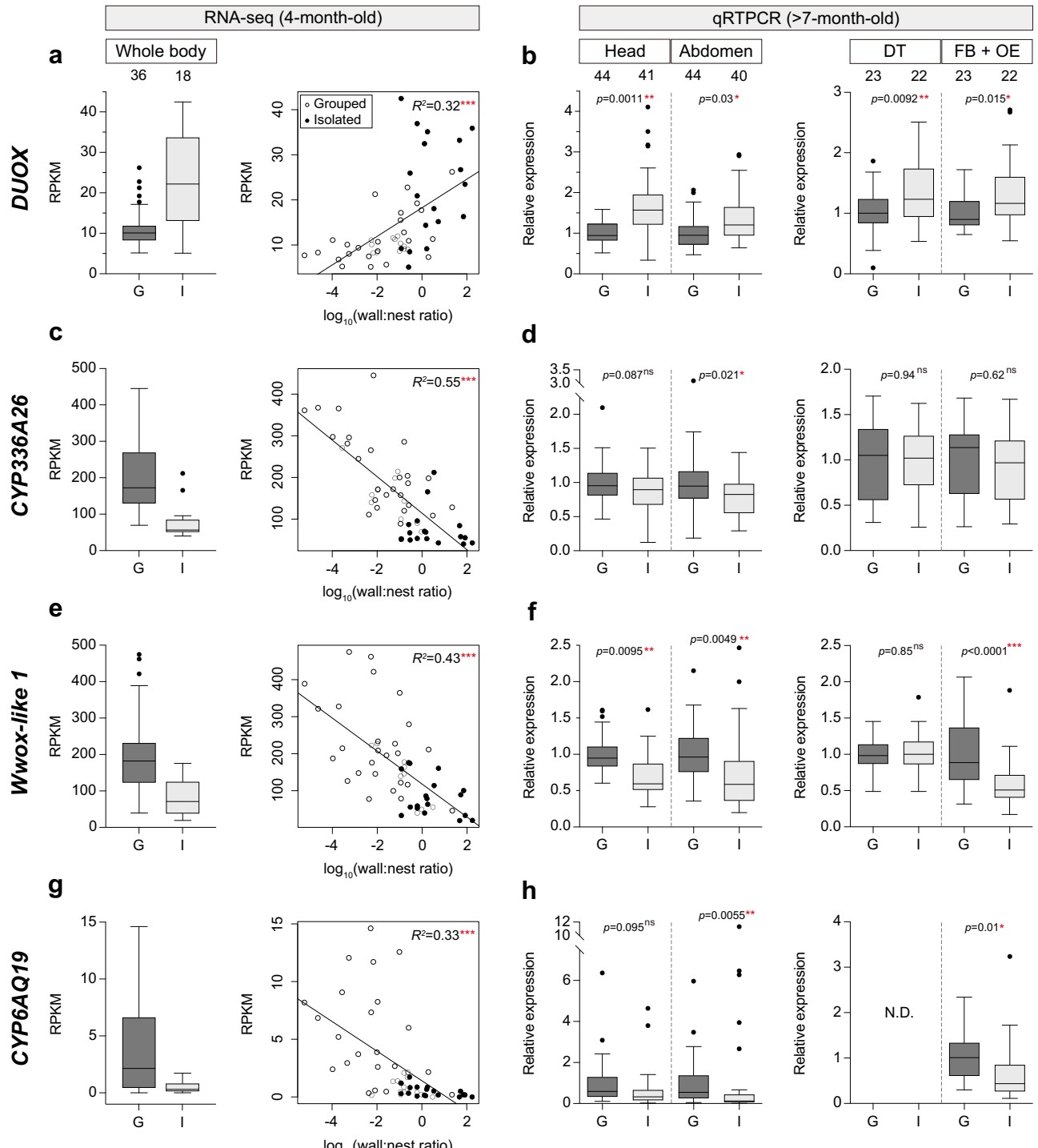

**Fig. 2 | Expression profiles of genes implicated in oxidoreductase activity in grouped and isolated ants. a**, **c**, **e**, **g** RPKM in grouped (G, dark gray, $n = 36$) and isolated ants (I, light gray, $n = 18$) (left), and the correlation between RPKM and wall:nest ratio (right) for the four DEGs, *DUOX* ($p < 0.001$), *CYP336A26* ($p < 0.0001$), *Wwox-like 1* ($p < 0.0001$), and *CYP6AQ19* ($p < 0.0001$) with the highest *q*-values and absolute $\log_2$ fold changes in gene ontology term; oxidoreductase activity in module 1. *R*-squared value from a simple linear model. **b**, **d**, **f**, and **h** Relative expression levels of *DUOX, CYP336A26, Wwox-like 1*, and *CYP6AQ19* in grouped (G, dark gray) and isolated (I, light gray) ants in the head ($n = 44$ for grouped ants and $n = 41$ for isolated ants) and abdomen ($n = 44$ for grouped ants and $n = 40$ for isolated ants) in the left panel, and digestive tract (DT, $n = 23$ for grouped ants and $n = 22$ for isolated ants) and fat body with oenocyte (FB + OE, $n = 23$ for grouped ants and $n = 22$ for isolated ants) in the right panel. Sample sizes (identical for the four genes) are given above graphs. Boxes and median lines represent inter-quartile range and median values, and whiskers represent minimum and maximum values of data within 1.5-fold of the inter-quartile range. Points indicate outliers. *P*-values are given in Supplementary Code 1. The significance of correlations between the RPKM and the wall:nest ratio (**a**, **c**, **e**, **g**) was tested with GLMMs and subsequent ANOVA tests, and the effects of treatment on relative gene expression (**b**, **d**, **f**, **h**) were tested with GLMMs and subsequent ANOVA tests with Tukey's post-hoc tests, *$p < 0.05$, **$p < 0.01$, ***$p < 0.001$; ns, not significant; N.D. not detected.

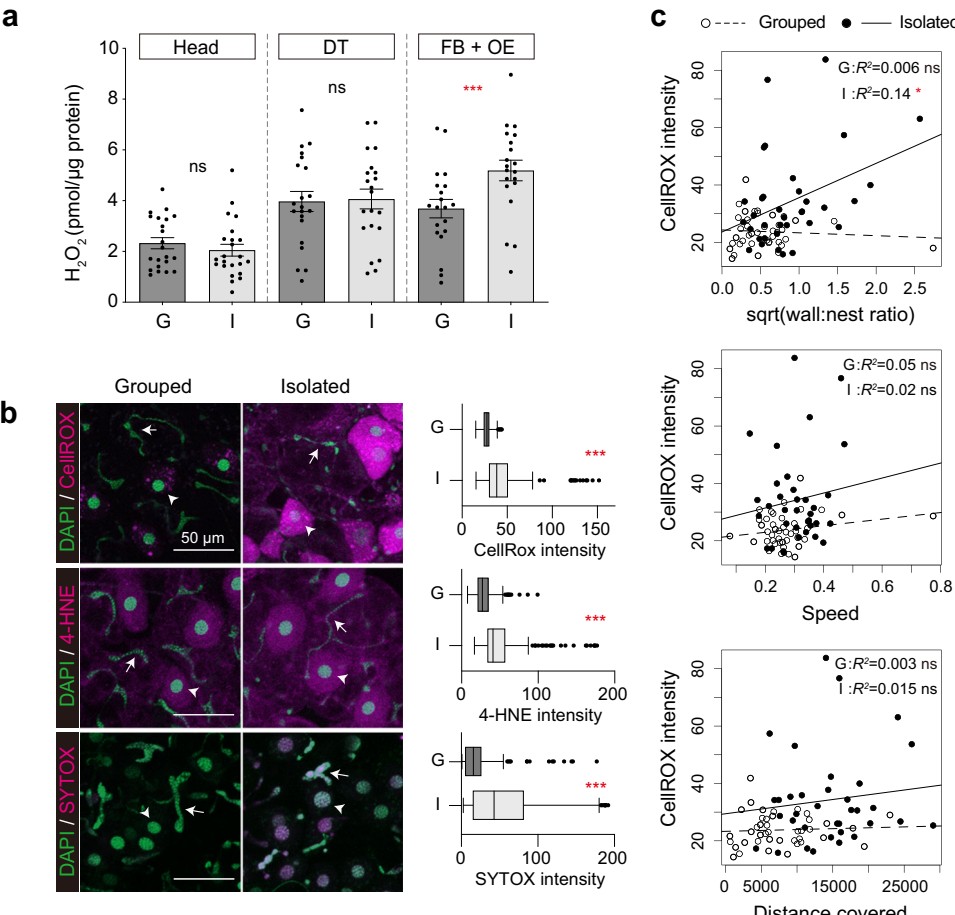

**Fig. 3 | ROS accumulation in the fat body and oenocyte under social isolation.**
**a** Quantification of hydrogen peroxide ($H_2O_2$, mean ± SEM with all data points) levels in the head (ctrl: $n = 22$, Mel: $n = 23$, $p = 0.44$), digestive tract (DT, ctrl: $n = 21$, Mel: $n = 21$, $p = 0.92$), and fat body with oenocyte (FB + OE, ctrl: $n = 20$, Mel: $n = 20$, $p < 0.001$) for grouped (G, dark gray) and isolated (I, light gray) ants.
**b** Representative images of oxidative stress markers in oenocytes and trophocytes (the main cell-type of the fat body) in grouped and isolated ants (left). CellROX (top), 4-HNE (middle), and SYTOX (bottom) are shown in magenta, and nuclei in green. Representative nuclei are labeled with white arrowheads for oenocytes, and white arrows for trophocytes. Scale bar: 50 μm. Quantification of CellROX (top, G: $n = 204$, I: $n = 214$, $p < 0.0001$), 4-HNE (middle, G: $n = 206$, I: $n = 269$, $p < 0.0001$) and SYTOX (bottom, G: $n = 299$, I: $n = 242$, $p < 0.0001$) in the oenocytes of grouped (G,

dark gray) and isolated (I, light gray) ants. Boxes and median lines represent inter-quartile range and median values, and whiskers represent minimum and maximum values of data within 1.5-fold of the inter-quartile range. Points indicate outliers.
**c** Relationship between ROS levels quantified with CellROX intensity and the behavioral parameters, wall:nest ratio (grouped: $p = 0.48$, isolated: $p = 0.021$), speed (grouped: $p = 0.085$, isolated: $p = 0.47$), and distance covered (grouped: $p = 0.66$, isolated: $p = 0.51$) in grouped (open circle with dashed line, $n = 41$) and isolated (filled circle with solid line, $n = 35$) ants. $R$-squared value from a simple linear model. GLMMs and subsequent ANOVAs with Tukey's post-hoc tests were used for (**a**), and Mann–Whitney U-tests were used for (**b**), and correlations between behavioral parameters and CellROX intensity (**c**) were tested with GLMMs and subsequent ANOVAs. *$p < 0.05$, ***$p < 0.001$; ns, not significant.

---

melatonin treatment fully rescued the pattern of space use of the isolated workers, which did not behave significantly differently from non-treated grouped ants ($p = 1.0$ in Fig. 4d). Overall, these rescue experiments suggest that the accumulation of ROS in the fat bodies and oenocytes of socially isolated ants contributes to the detrimental effects of social isolation on lifespan and behavior.

## Discussion
Our study reveals that the social isolation-induced changes in physiology and behavior are mediated by changes in oxidoreductase activity. Social isolation led to changes in the expression of key oxidoreductase genes and the accumulation of ROS, especially in the fat body and oenocytes. A similar link between social isolation and oxidative stress has recently been reported in other animals. In *Drosophila*, chronic social isolation causes sleep loss, increases feeding, and alters the expression of genes implicated in the oxidation-reduction pathway[12]. Impairment of the antioxidant system and increased oxidative stress has also been reported in the central nervous system and peripheral tissues of socially isolated rodents[66–69]. Together with our

results, this suggests that increased oxidative stress may detrimentally affect the physiology and behavior of socially isolated animals in general.

Our results showed that social isolation-induced ROS accumulation occurs primarily in the fat body and oenocytes, the hepatocyte-like cells of insects. In *Drosophila*, fat body and oenocytes share liver-like functions[59]. Aging results in increased ROS production in adult oenocytes in *Drosophila*[70]. Similarly, in rodents, the livers of older individuals show increased oxidative stress and decreased detoxification capacity[71,72]. In insects, oenocytes are involved in lipid metabolism and the biosynthesis of cuticular hydrocarbons and pheromones[59,73]. Cell-type-specific ribosome profiling of oenocytes revealed that with aging there was a decrease in the expression of genes involved in oxidative phosphorylation, fatty acid metabolism, and peroxisomal enzymes in *Drosophila*[70]. Another recent study suggested that impairment of peroxisome function induces the release of pro-inflammatory factors from oenocytes, resulting in cardiac dysfunction in old flies[74]. Thus, oenocytes seem to play a key role in the regulation of energy

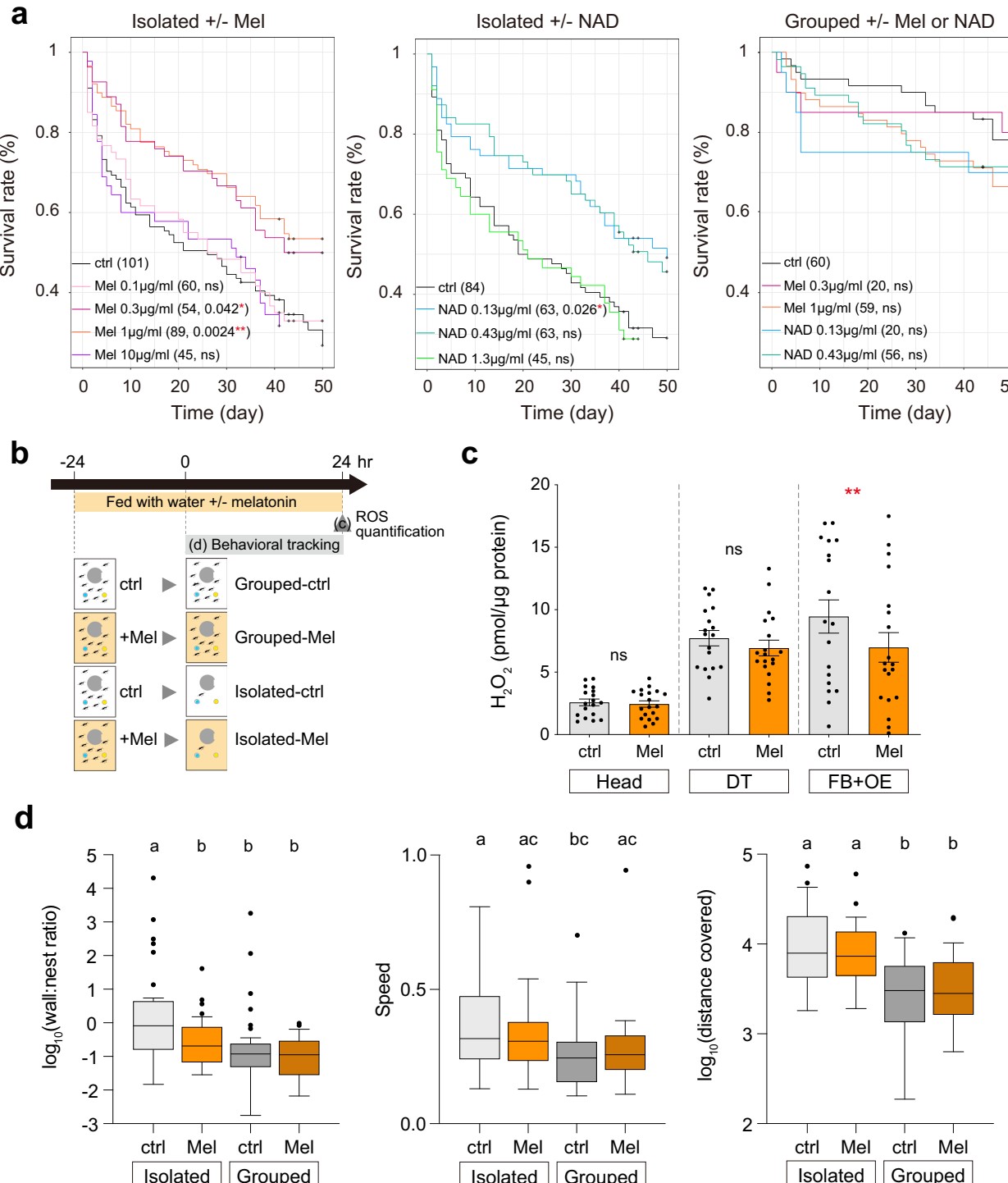

**Fig. 4 | Suppression of ROS prevents early death and abnormal behavior under social isolation. a** Survival curves for isolated ants given a control (ctrl) treatment or treated with different doses of antioxidant melatonin (Mel, left) or antioxidant nicotinamide adenine dinucleotide (NAD, middle). Sample sizes and $p$-values are shown in the legend. Survival curves for control (ctrl), melatonin-, or NAD-treated grouped ants are shown on the right. **b** Scheme of the experimental design to test the role of antioxidant treatment on ROS levels (**c**) and behavioral parameters (**d**) in grouped and isolated ants. Symbol: 'Ant' by Jacob Eckert from the Noun Project. **c** Quantification of hydrogen peroxide ($H_2O_2$, mean ± SEM with all data points) in the head ($p = 0.95$), digestive tract (DT, $p = 0.4$), and fat body with oenocyte (FB + OE, $p = 0.0051$) for control (ctrl, light gray, $n = 18$) and melatonin-treated (Mel, orange, $n = 19$) workers under social isolation. **d** Box plots of the ratio of time spent near the wall to time spent in the nest, speed (cm/frame), and distance covered (cm/day) of control (ctrl, light gray) and melatonin-treated (Mel, orange) workers in isolated (ctrl: $n = 38$, Mel: $n = 36$) or grouped (ctrl: $n = 43$, Mel: $n = 40$) conditions during 24 h of tracking. Boxes and median lines represent inter-quartile range and median values, and whiskers represent minimum and maximum values of data within 1.5-fold of the inter-quartile range. Points indicate outliers. $P$-values are given in Supplementary Code 1. The effect of treatment on ROS production (**c**) and behavior (**d**) was tested using GLMMs and subsequent ANOVAs with Tukey's post-hoc tests (**c**) and Bonferroni post-hoc tests (**d**). *$p < 0.05$, **$p < 0.01$; ns, not significant. Groups differing significantly ($p < 0.05$) are marked with different letters in (**d**).

metabolism, peroxisome, systemic inflammation, and ROS production, all of which are implicated in aging.

Oxidative stress has been linked with aging and senescence in a variety of organisms[75]. Because of the large difference in lifespan between queens and workers, social insects represent a promising model system to study the plasticity of aging and senescence[76–78]. A comparative study of 20 genes commonly involved in managing oxidative damage in queens and workers of four social insects (a termite, two bees, and an ant) indicated that oxidative stress is a significant factor in senescence but that its manifestation and antioxidant defense mechanisms differed among species[77]. In this study, the authors analyzed different tissues in the four species investigated. Our study showed that the effect of social isolation on the expression of genes implicated in the production and detoxification of ROS differs between tissues, which may explain some of the interspecific differences observed by Kramer et al.[77].

Social isolation impairs mammalian immune systems[8,79] and causes the differential expression of immune system-related genes in the ant *T. nylanderi*[36]. However, social isolation did not affect immune system-related genes in our experiment. The contrasting results of the two ant studies may result from the analysis of different tissues (whole body in our case and brain for *T. nylanderi*), or interspecific differences in the consequences of stress on the immune system. Our experiments revealed that pathways other than oxidoreductase activity were also affected by social isolation (e.g., transporter activity and modified amino acid binding). Some of these changes may be due to isolated ants being unable to perform trophallaxis with nestmates or to the isolation-associated impairment of digestion[31].

To demonstrate that changes in oxidoreductase activity and ROS accumulation causally underpin the reduction in lifespan associated with social isolation we administered melatonin, a hormone primarily known for its regulation of vertebrate circadian rhythms[80]. In invertebrates, melatonin does not seem to regulate circadian rhythm, but has radical scavenger effects and antioxidant properties[81]. The administration of melatonin resulted in increased lifespan of isolated but not grouped ants. The administration of another antioxidant (NAD) similarly only increased lifespan of socially isolated ants, demonstrating that in our experiments, antioxidants had positive effects only for individuals suffering from social isolation (if anything there was a tendency for the two antioxidants to decrease the lifespan of grouped ants). Importantly, the supplementation of melatonin had no significant effect on the speed of isolated ants, or on the distance that they covered, but did significantly reduce the time spent near the wall. Treatment with melatonin fully rescued the pattern of space use of isolated workers but did not affect the space use of grouped workers, again demonstrating that the positive effect of antioxidants was restricted to individuals suffering from social isolation. Our experiments also demonstrated that the isolated workers which spend the most time near the wall were the workers that exhibited the highest levels of ROS suggesting that change in space use is a good marker of the degree of stress induced by social isolation. Social isolation has also been reported to affect space use in mice, where isolation reduced exploratory behavior in the central zones and increased the time spent in peripheral regions of enclosures[82,83]. Although our experimental setup differs from the open field test commonly used to measure anxiety-related behavior and exploration in rodents[84], it is interesting to note that ants may have similar behavioral response to mice, staying near box edges when isolated.

Our results also revealed that the melatonin treatment had a small and non-significant effect on ROS levels in the head and digestive tract but was effective in clearing ROS from the fat body and oenocytes. These results are interesting given that the increased expression of key oxidoreductase genes and the accumulation of ROS induced by social isolation occurred primarily in the fat body and oenocytes. Overall, these rescue experiments demonstrate that the accumulation of ROS

in the fat bodies and oenocytes of socially isolated ants causally underpins the detrimental effects of social isolation on lifespan and behavior. In conclusion, our findings have important implications for understanding how social perturbations affect the expression of genes, and consequently the health and lifespan of social organisms. Our study suggests that oxidative stress causally mediates the negative effects of social isolation on behavior and lifespan and that the oenocytes and fat body play a key role in this process.

## Methods

This study did not require any ethical approval.

### Ants

*Camponotus fellah* colonies were initiated from queens collected after a mating flight in March 2007 or 2010 in Tel Aviv, Israel. The ants were reared in an incubator (NIPPON MEDICAL & CHEMICAL INSTRUMENTS CO., LTD) under controlled conditions (12:12 LD, 30 °C, 60% RH). For all experiments, we used minor workers (body size <8 mm) from queenright colonies that each had one queen and approximately 1000 workers. To determine their age, we painted all newly-eclosed workers monthly with a unique color code. We previously found that isolation reduced lifespan independently of age[31] and used 4-month-old workers for RNA-seq analysis because older workers contain more formic acid which hinders measurement of RNA concentration. We used >7-month-old workers for other experiments including qRTPCR (Fig. 2b, d, f, h), ROS quantification (Fig. 3), survival test, ROS quantification, and behavioral analysis with antioxidants (Fig. 4).

Ants were separated from their colony and reared in groups of 10 (grouped) or alone (isolated) in a plastic box (105 × 87 mm) containing food (made from honey, eggs, and vitamin tablets), water, and a light-shielded nest box (28 mm diameter for isolated, and 52 mm diameter for grouped treatments). For each experiment (except data in Fig. 3b), the experimental colonies were created using 2–4 independent colonies of origin (Fig. 1 RNA-seq: 3 colonies; Fig. 2 qRTPCR for parts: 3 colonies; Fig. 2 qRTPCR for tissues: 2 colonies; Fig. 3a ROS quantification: 3 colonies; Fig. 3c ROS and behavioral quantification: 3 colonies; Fig. 4a Survival with Mel: 4 colonies; Fig. 4a Survival with NAD: 3 colonies; Fig. 4a Survival in grouped: 4 colonies; Fig. 4c ROS quantification: 3 colonies; Fig. 4d Behavioral quantification: 3 colonies). Colony-of-origin effects were accounted as a random factor in each experiment, except for the immunohistochemistry data in Fig. 3b where all individuals originated from the same colony.

### RNA extraction, library preparation, and sequencing

After 24 h of behavioral tracking, each ant was flash-frozen in liquid nitrogen and total RNA was extracted from the whole body with Trizol (Invitrogen, 15596026) and cleaned with RNeasy plus micro kit (QIAGEN, 74034) following the manufacturer's instruction for RNA-seq analysis. After precipitation, RNA quality was assessed with a nanodrop 1000 spectrophotometer (Thermo Fisher Scientific) and a Bioanalyzer (Agilent). Using 200 ng of total RNA per sample as template, cDNA libraries were constructed with a Kapa stranded mRNA-seq kit (KAPABIOSYSTEMS, KK8421) using PentaBase indexed adaptors (labgene scientific). Libraries were quantified with Qubit 2.0 Fluometer (Invitrogen) and then qualified with Bioanalyzer. When primer dimers were detected, they were removed via bead purification (AMPure XP, A63880, Beckman Coulter). cDNA libraries were sequenced using strand-specific single-end sequencing of 100 bp reads with an Illumina HiSeq machine (HiSeq2500).

### Reference genome assembly and annotation

Three Oxford Nanopore sequencing libraries were prepared from high molecular weight genomic DNA using the RAD002 kit and sequenced on a MinION Mark1-B on three rev-D R9.4.1 flowcells. Reads were basecalled using Guppy v2.1.3 (Oxford Nanopore Technologies) and

assembled using wtdbg2[85]. The assembly was then polished with Nanopore long reads using Medaka v0.6 (https://github.com/nanoporetech/medaka), then with Illumina reads first using Pilon[86], and then using variants called by freebayes (https://github.com/ekg/freebayes) with variant quality scores greater than 30.

Candidate annotations were generated using the hint guided gene predictors Augustus[87] and SNAP[88], genome-guided RNA-seq assembly from the Trinity assembler[89] and PASA[90] refinement tool, and liftover of NCBI RefSeq protein annotations from *Camponotus floridanus* using GenomeThreader[91]. Augustus and SNAP were run through the MAKER pipeline[92] with protein homology evidence from the NCBI RefSeq *Nasonia vitripennis*, *Apis mellifera*, *Camponotus floridanus*, and *Ooceraea biroi* proteomes and RNA-seq alignments. MAKER was run first using pre-packaged ab initio models from *Nasonia vitripennis* for Augustus and from *Apis mellifera* for SNAP, then again with the same evidence but with ab initio models retrained from the first round. All candidate gene models were then combined using EvidenceModeler[93] with the above described transcript and homologous protein alignments used as evidence.

## Phylogenic analysis

To determine the identity of the two *CYP* genes that were differently expressed between grouped and isolated ants, we performed a phylogenic analysis of CYP clan 3, to which the two CYPs belong, using the amino acid sequences obtained from NCBI using a blast search with the DEGs as queries (Supplementary Fig. 4a, Supplementary Code 6 and 7). Amino acid sequences of Hymenopteran CYP proteins in Supplementary Fig. 4b, Supplementary Code 8 and 9 were obtained from Nelson's cytochrome P450 website[94] and annotated transcript sequences from the reference genome of *Camponotus fellah*. Amino acid sequences were aligned using MAFFT version 7.490 in the E-INS-i mode[95]. Gene trees were reconstructed using RAxML version 8.1.15 with the PROTCATLG model, i.e. the LG substitution matrix and the CAT approximation for among site rate variation, and 100 bootstrap iterations[96]. Trees were visualized using Figtree ver 1.4.0 (http://tree.bio.ed.ac.uk/software/figtree/).

## Gene expression analysis

Sequenced reads were mapped to annotated transcript sequences using the BWA-MEM program implemented in MASER pipeline[97], whereby transcript expression levels were estimated in terms of reads per kilobase of exon per million mapped reads (RPKM). cDNA sequences were annotated with blastx against the *Drosophila melanogaster* database (ver. 6.21) and the *Camponotus floridanus* database (ver. 7.5).

## Differential gene expression analysis and gene ontology analysis

Differential expression analysis was assessed using edgeR in R (ver.3.6.1) using RPKM values normalized with the TMM (trimmed mean of M-values) method. To exclude the DEGs with outliers, we calculated the mean RPKM value from grouped (42 samples) and isolated (18 samples) treatments, and the ratio of RPKM between isolated and grouped treatment (referred as ratio with all samples). We next calculated the mean RPKM value in isolated and grouped treatments excluding the minimum and maximum values of RPKM in each social treatment, and then calculated the ratio of RPKM without the minimum and maximum values between isolated and grouped treatment (referred as ratio without max/min). We then excluded the genes with an absolute $\log_2$ ratio (ratio with all samples/ratio without max/min) >1 as outliers. For the volcano plot, genes with mean RPKM > 1.5 and median RPKM > 0 were plotted. Differentially expressed genes (*q*-value < 0.05 and absolute $\log_2$ fold change ≥1) were shown in a heatmap using the function 'clustergram' in MATLAB 2020a. Gene Ontology enrichment analyses were performed using a

Benjamini–Hochberg FDR *p*-value correction in gProfiler2 (Ensemble 103; https://biit.cs.ut.ee/gprofiler/gost).

## Weighted gene co-expression network analysis

The weighted gene co-expression network analysis (WGCNA) was performed using RPKM values as previously described[98]. Module eigengenes represent the gene expression profiles of each module[99]. Genes with insufficient variance (standard deviation ≤1) and low expression (median ≤1) were excluded. A soft-threshold power of 7 was chosen to build a scale-free topology using a signed hybrid network. We set the minimum module size to 40, and merged modules with correlation coefficients greater than 0.8 (mergeCutHeight = 0.2). To assess the correlation of modules to behavioral parameters, Pearson correlation coefficients were computed between module eigengenes and behavioral parameters.

## Quantitative RT-PCR

RNA was extracted with TRIzol from the heads or abdomens of single ants[100], and from the digestive tracts or fat body with oenocytes from pools of 3 ants to ensure a sufficient quantifty of RNA. cDNA was synthesized from 200 ng of total RNA with a PrimeScript RT Reagent Kit with gDNA Eraser (Takara, RR047). qRTPCR was performed with Takara SYBR Premix Ex Taq II (Tli RNaseH Plus) with using QuantStudio 5 (Thermo Fisher). Primers for qRTPCR are listed in Supplementary Table 1. All primers had similar PCR amplification efficiency (~2.0). The most stable reference genes (*ef1a* and *rp2*) and gene normalization factors were determined by geNORM with Biogazelle (qbase). The relative expressions of target genes were scaled against average values with the Biogazelle software and then normalized against the mean values in the grouped treatment for each body part/tissue.

## ROS measurement

Heads, digestive tracts, and fat bodies with oenocytes from three workers were transferred in 100 μl of 0.1 M phosphate-buffered saline (PBS). Samples were homogenized and centrifuged to remove debris. Protein concentration was determined with the BCA assay kit (Thermo Fisher, 23225). Homogenized tissue extract for 10 μg protein was diluted to 50 μl with PBS, and the level of hydrogen peroxide was measured by the addition of 50 μl reaction buffer (50 μM Amplex Red (Invitrogen, A12222); 0.1 μM horseradish peroxidase (Wako, 169-10791) in PBS). Fluorescence intensity was measured with a fluorescence microplate reader (Infinite 200Pro Mplex, Tecan) using an excitation wavelength of 530 nm and an emission wavelength of 590 nm. A standard curve was plotted with a range of $H_2O_2$ dilutions and used to determine the amount of $H_2O_2$ per sample, which was then normalized to the amount of protein. For the ROS measurement of grouped and isolated ants in Fig. 3a, we either isolated ants or placed them in groups of 10 for 24 h before dissection. For the ROS quantification of isolated ants given either water or water with melatonin in Fig. 4c, pretreatment was performed as described in "Antioxidant feeding". ROS levels in the fat bodies with oenocytes were visualized using CellROX Deep Red reagent (Life Technologies, C10422) following the manufacturer's protocol. The fat bodies with oenocytes were dissected in ant saline[101] and incubated in 1 ml ant saline with 5 μM CellROX Deep Red for 45 min at 29 °C. The samples were washed three times with PBS and fixed with 4% paraformaldehyde (PFA) in PBS for 15 min at room temperature. Samples were then washed three more times with PBS. In Fig. 3c, CellROX staining in the fat body with oenocytes was performed after 24 h of behavioral tracking in the isolated or grouped condition.

## SYTOX Green necrosis assay

Necrosis was detected with SYTOX Green nucleic acid stain (Invitrogen, S7020). The fat bodies with oenocytes were dissected in ant saline and incubated in 5 μM SYTOX Green in 4% PFA for 30 min at room

temperature on a shaker. Samples were then washed three times with PBS at room temperature on a shaker.

## Immunohistochemistry

Immunohistochemistry for the fat bodies and oenocytes was performed as described previously[100], using anti 4-Hydroxynonenal (4-HNE) antibody (1:100, ab46545, Abcam) overnight at 4 °C, and Alexa 488-conjugated anti-rabbit secondary antibody (1:1000, A-21206, Invitrogen, in PBS with 0.2% TritonX-100 and 5% normal donkey serum) for 2 h at room temperature. Samples were then washed three times with PBS with 0.2% TritonX-100 at room temperature on a shaker.

## Image acquisition and quantification

Samples were mounted with SlowFade Diamond Antifade Mountant with DAPI (Invitrogen, S36964) after the final wash and imaged on a Zeiss LSM980 confocal microscope (Carl Zeiss) with a Plan-APOCHROMAT 20x/0.8 or 40x/1.4 objective (Carl Zeiss) and ZEN3 software. All images for each reagent (CellROX, 4-HNE, or SYTOX Green) were captured in the same imaging parameters including the exposure time, gain, or laser power. Pixel intensities of Z stacks (merged slices) were used for quantification of CellROX Deep Red, 4-HNE, or SYTOX Green staining. The mean of summed pixel intensities in the cell body (CellROX and 4-HNE) or nucleus (SYTOX Green) of each oenocyte was calculated with Fiji (ImageJ) software and used for statistical analysis in Fig. 3b.

## Antioxidant feeding

Grouped or isolated workers were placed in an insect breeding dish (Bio Medical Science) containing a light-shielded nest box, food, and water. Groups of 10–15 workers were placed in insect breeding dishes with or without antioxidants in the water for 24 h. Workers were then placed into insect breeding dishes, either alone or in groups of ten and used for the survival assay (Fig. 4a), ROS quantification (Fig. 4c), and the behavioral assay (Fig. 4d). We tested three antioxidants, melatonin (Mel, M5250, Sigma-Aldrich), nicotinamide adenine dinucleotide (NAD, N0632, Sigma-Aldrich) and lipoic acid (Sigma-Aldrich, T5625) which clear ROS from *Drosophila* guts[50]. Because lipoic acid led to high mortality, we used those treated with melatonin and NAD for the analysis.

## Survival analysis

Survival of the grouped and isolated ants with and without antioxidant feeding was monitored every day after the 24-h pre-treatment. The ants had access to a water supply with or without antioxidants that was replenished daily. All boxes were kept at 30 °C under 12 h light-12 h dark cycles. We analyzed the survival of ants with a mixed effects cox regression model in R (function coxme from package coxme in R 3.6.1) with one fixed factor (antioxidants) and one random factors (colony of origin) for isolated, or two random factors (colony of origin and box id) for grouped ants with Dunnet post-hoc tests.

## Behavioral tracking and data processing

Behavioral tracking and the data processing were performed as described previously[31,41,102]. Ants were kept for 24 h in the grouped or isolated condition in plastic boxes (105 × 87 mm) containing food, water, and a light-shielded nest. Behavioral tracking of the workers used for RNA-seq analyses was performed with tracking systems described previously[31,41]. The behavioral tracking of the workers used for ROS quantification (Fig. 3c) and antioxidant treatment (Fig. 4d) were performed with a monochrome high-resolution camera (RMV-29050, illunis) and enlarging lens (Color-Skopar SL Aspherical 28 mm, Voigtländer, or 14–24 mm F2.8 DG HSM, SIGMA). To automatically infer space use, we defined four regions: near the wall, a

food area, a water area, and the arena (Supplementary Fig. 1a), and then calculated the total duration spent in each region of interest for 24 h. We calculated the duration of time spent in the nest as the sum of differences between each time of entry and time of exit of the nest, as done previously[31], and manually confirmed entry and exit times with the video feed; Figs. 3c and 4d. The time point of each nest entry and exit for each ant in Figs. 1a, b, 3c, and 4d are included in Source Data 1. The duration spent in each region was weighted by the total number of frames in which the individual was detected during the 24 h. The distance covered was estimated as the sum of Euclidean distances between all subsequent positions as previously reported[31], and the speed was calculated from the total distance covered and the number of timesteps when individuals were moving. Among the 60 samples in the RNA-seq experiment, the six grouped samples with a detection rate <10% were excluded from the behavioral analysis in Figs. 1a, b, c, g, 2a, c, e, g. For behavioral tracking with melatonin, pre-treatment was performed as described in "Antioxidant feeding". After 24 h of antioxidant pre-feeding, each worker or groups of 10 workers were placed in a different box with or without antioxidant in the water supply.

## Statistical analyses

To compare the behavior of grouped and isolated ants, we fitted a GLMM with one fixed factor (social treatment) and two random factors (box id and colony of origin) in Fig. 1a and Supplementary Fig. 1b. using R (R 3.6.1). Correlations between the time spent near the wall and in the nest in Fig. 1b, or between the RPKM and wall:nest ratio in Fig. 2a, c, e, g were tested with a GLMM with two random factors (box id and colony of origin). For the gene expression analysis in Fig. 2b, d, f, h, we fitted a GLMM with two fixed factors (social treatment and body parts/tissues) and two random factor (box id and colony of origin) in body parts, and one random factor (colony of origin) in tissues. For the ROS analysis in Figs. 3a and 4c, we fitted a GLMM with two fixed factors (social treatment (Fig. 3a) or antioxidant treatment (Fig. 4c), and tissues) and one random factor (colony of origin). Correlations between the behavioral parameters and ROS intensity were tested using a GLMM with one random factor (colony of origin) for isolated, and two random factors (box id and colony of origin) for the grouped treatment in Fig. 3c. Signal intensities of CellROX, 4-HNE, and SYTOX Green were compared between the oenocytes of grouped and isolated ants using Mann–Whitney U tests in Fig. 3b. For the behavioral analysis in Fig. 4d, we fitted a GLMM with two fixed factors (social treatment and antioxidant treatment), and two random factors (box id and colony of origin). Tukey's correction (Figs. 2b, d, f, h, 3a, and 4c) and Bonferroni correction (Fig. 4d) were applied to correct for multiple testing. We first tested whether the data were normally-distributed with the Kolmogorov–Smirnov test and diagnostic qqPlots. When the residuals were not normally distributed, we applied the $\log_e$ or square root transformation, which normalized the residuals. Sample numbers indicate the number of biologically independent samples in each experiment. Statistical values for data in Figs. 1–4 and Supplementary Fig. 1b are listed in Supplementary Code 1.

## Reporting summary

Further information on research design is available in the Nature Portfolio Reporting Summary linked to this article.

# Data availability

Datasets for genome sequence, cDNA lists, RNA sequence, and images used for the quantification of ROS markers are deposited in the repositories listed in Supplementary Data 2. All other relevant data supporting the findings of this study are included within the article as Source data and Supplementary Information. Source data are provided with this paper.

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

## Acknowledgements

We thank A. Hefetz for collecting queens, and M. Sakurai, K. Iwata, Y. Iijima, and J. Itoh for help with animal keeping, gene expression, and immunohistochemistry analyses. We are grateful to O. Riba-Grognuz for technical help with RNA-seq analysis, R. Futahashi for his help in RNA-seq data analysis, D. Mersch for modifying the program of behavioral tracking, and R. Benton for commenting on the manuscript. This work was funded by grants from AMED under Grant Number JP18gm6110014, JSPS KAKENHI Grant Number JP21H05297 to A.K., and the Swiss NSF Grant 310030B_176406 and 310030_200437, and ERC Grant resiliANT 741491 to L.K. The funders had no role in the study design, data collection and analysis, decision to publish, or preparation of the manuscript.

## Author contributions

A.K. planned and directed this study, performed experiments, and wrote the manuscript. M.T. conducted the WGCNA analysis, and P.S.W. and S.A. performed the behavioral data analysis and contributed unpublished essential data. A.C. modified the behavior tracking system. C.S. performed the library preparation of RNA-seq. E.P. performed the transcriptome and phylogeny analysis. C.L.M. performed sample preparation of DNA-seq and S.K.M. performed genome assembly and annotation of Camponotus fellah. T.K. also performed sample preparation of DNA-seq and wrote the manuscript. L.K. directed the project, analyzed the data, and wrote the manuscript.

## Competing interests

M.T. is an employee of Mitsubishi Tanabe Pharma America. P.S.W. is currently an employee at Human Resocia. S.K.M. is currently an employee at Oxford Nanopore Technologies and owns stock in said company. The remaining authors declare no competing interests.
