## [Peer Review File · Nature Communications]

Social isolation shortens lifespan through oxidative stress in antsReviewers' Comments:

Reviewer #1:

Remarks to the Author:

Social animals are known to be stressed when kept in isolation. This experimental study analyses the effect of social isolation on workers of the ant *Camponotus fellah*, showing an increase in reactive oxygen species in isolated workers. The production of ROS is reduced when workers are treated with melatonin, experimentally rescuing workers from oxidative stress and stress related behaviours. This rescue has been shown before in the solitary *Drosophila*, indicating that the link between ROS production, stress and melatonin is not linked to sociality. The strength of the study is demonstrating experimentally that short isolation (1 day) leads to ROS production in a social insect, which in turn shortens the lifespan of ant workers. On the other hand, that social animals including ants are stressed by isolation is not novel and also the behavioural analyses only investigating the time close to the wall / in the nest, is not very revealing and detailed. While the results are well written, the discussion could clearly be improved by a more critical, multifaceted reflection of the findings.

Introduction

In general the introduction starts with a focus on human studies, and then introduces social insects, as useful models to study the mechanistic effects of social isolation. This framing suggests that the mechanisms underlying the negative effects of social isolation are the same in humans, social mammals and insects. However, this might not be the case and at least a critical reflection or citations of studies showing mechanistic similarities would improve the introduction.

Line 44 References strongly focusses on human studies, even though it is a quite general statement

Results

Line 76-78 The behavioural observations are not very detailed indeed the authors rather report the location (next to the wall, in the nest) instead of the behaviours. The locational shift can easily explained by isolated workers having to leave the nest for food or in search for nestmates. Whether this "behaviour" is really a stress behaviour remain unclear.

Line 77 How long where ants isolated? Only 24h? Why this rather short social isolation Time?

Line 80 If you only record to location (next to wall, in nest) is it not unavoidable to find a negative correlation?

Methods

Line 369 Why the different ages for the ants in the RNA seq and experimental studies? Could that effect the interpretation of the results

Line 377 Specify the differences between ROS quantification in Fig 3a or Fig 4b

Line 418 Which amino acid sequences were analysed? What was the reason for this phylogenetic analyses? Please clarify.

Line 533 For how long where the ants isolated? It remains unclear, in which experiment, which isolation regime was done.

Reviewer #2:

Remarks to the Author:

Koto and colleagues describe behavioral, genetic, cellular, and molecular effects of social isolation on the carpenter ant, *Camponotus fellah*. This work builds upon previous studies that have demonstrated negative effects of social isolation in both social and non-social taxa, including decreases in lifespan. However, uncovering the molecular and cellular underpinnings of aging phenotypes has been challenged by variable results between studies – which in many cases may reflect true differences between experimental conditions and taxa. Relatively few studies go beyond correlation to demonstrate causal effects of the genes or molecules identified. By following up on changes in gene expression in socially isolated workers, Koto et al. identify tissue-specific ROS damage in socially-

isolated ants that they rescue pharmacologically. The authors confirm an earlier study that showed socially isolated workers spend less time near the nest and more time near the walls of an arena – an atypical behavior likely indicating stress. Their analysis of differential gene expression identifies an enrichment in genes related to oxidoreductase activity, consistent with ideas that reactive oxygen species (ROS) are causally involved in many senescence-associated phenotypes. Using RNA-seq and qRT-PCR, they find significant differences in several ROS-associated genes, notably DUOX, an NADPH-oxidase, and three genes involved in detoxification. The authors next confirmed higher levels of ROS in fatbodies and oenocytes, but no significant difference in the heads and digestive system of the ants. This tissue-specificity is consistent with other studies, several of which show lack of oxidative damage in the nervous system with aging. Koto and colleagues conclude the paper with the most compelling result – they significantly increase lifespan in socially isolated ants by feeding them antioxidants. This finding demonstrates quite convincingly that oxidative stress is causally involved in the shortened lifespans of socially-isolated ants. This work makes an important contribution to our understanding of aging in both eusocial and non-eusocial taxa and provides specific molecular pathways that should be examined in other systems. The findings are novel and of broad interest to researchers focused on the physiology of aging, social insects, and mechanisms involved in senescence.

However, before publication, several aspects of the manuscript require clarification or additional details. I have one area of significant methodological concern, regarding the quantification of fluorescence. The intensity of fluorescence obtained from a confocal microscope can be strongly influenced by imaging parameters, such as gain and offset. Generally, quantifying intensity can be problematic, especially if all images are not acquired under the same conditions. The methods section neither describes the imaging parameters used nor whether they were consistent across treatments. While the data from the ROS quantification (Fig 3A) are convincing, I am not sure how to interpret the data presented in Fig 3B. If the images were not acquired under the same conditions, they could potentially be normalized to the DAPI intensity to allow comparisons between treatments. The authors should provide additional details and perhaps analysis to demonstrate that quantitative comparisons can be made between different images.

My other concerns and comments are minor, and I address these aspects line by line below. If all concerns are addressed, I strongly recommend publication for this interesting and important work.

Line 52: The authors should broaden the context of their work to include more references to aging in social insects outside of *Camponotus*. Notably studies on other ant species and bees would be useful to reference here.

Line 184: It is possible that the effects of antioxidants are not specific to social isolation, but could act to increase lifespan under conditions of high ROS/low antioxidant activity, which in this case was induced by social isolation.

Line 218: Change hydrocarbon to hydrocarbons.

Line 239: Is this supposed to be a new paragraph or just new sentence?

Line 254: As mentioned for line 184 above, it is possible that rescue effects of antioxidants in socially-isolated workers does not indicate that they only act during social isolation, but rather, exert an effect on lifespan only when under high ROS/low antioxidant conditions. Other stressors could similarly shorten lifespan but then be rescued by such a treatment, but that of course awaits further study.

Line 396: Why were 4- and 5-month old ants used for the RNA-seq analysis while 7-month olds used for all other aspects of the study? Given the importance of age on the phenotypes measured it would be useful to know why this was done and if there is any significant difference in gene expression between these two age groups.

Line 496: Even though the method is described in another paper, it would be helpful to briefly state the incubation times and temperatures for the primary and secondary antibodies.

Line 534: Please provide a brief description of the behavioral tracking assay, space allowing. Although the details can be found elsewhere, it would be very helpful to have a 1-2 sentence description of the tracking and analysis. It would be interesting to know if other behaviors than time in nest and time at the wall were tracked. If so, did these show a significant effect of treatment? Such data could appear in the supplement.

Figure 4: The yellow survival curves on the white background for the middle panel (4a) are hard to see. I would recommend selecting a different color.

Reviewer #3:

Remarks to the Author:

The manuscript is based on the experiment carried out with an ant *Camponotus fellah* and studies the effect of social isolation on the oxidative stress in the worker ants. The authors use different molecular, histological methods as well as bioassays to examine the role of the social isolation and the effect of ROS on the longevity in the ant workers. They claim that the release of free radicals is the cause of reduced life span in the isolation and feeding antioxidants will decrease the ROS-induced mortality in the social isolation.

While the topic is interesting in the ant world, all in all I feel that authors are unfortunately oversimplifying the results and the data collection for the claims is not done properly. Please see my comments below.

The way the experiments are carried out raises several concerns:

- Much of the conclusions are based on the RNA-sequencing data, which is based on the material originating from ants, which have been in isolation for 24h. At the same time, it is stated that social isolation leads to higher activity levels – workers are moving around more. One would argue that the higher levels of ROS in the organism are linked to the physical activity (which is known to generate ROS) rather than to the social isolation. Even if the higher activity levels are due to the ant worker being in isolation, the cause increased ROS is still physical activity.
- As the RNA-sequencing was carried out in 4-5 month old ants, and the rest of the studies in much older individuals (more than 7) it really is impossible to extrapolate any results over these age classes. Even, if the workers are relatively long-lived (to which I did not find any reference to in the text), one can expect certain age-related physiological changes taking place in the organism. It is known, that one of the ageing related effects is the accumulation of ROS-related damages. Hence, it is of course not a surprise, that feeding antioxidants can alleviate this effect.
- Alternative explanations for the life-span differences should be considered. It is well possible, that trophallaxis, which was not available for isolated ants can transfer certain molecules which are beneficial, the lack of access to this can be detrimental. And, while it is of course linked to isolation it is not a direct cause.
- It is a bit hard to follow the timeline and ages of the ants for different assays. The manuscript would benefit from the addition of a figure which highlights how the samples were taken and when the samples were taken for different analysis.
- Both melatonin and nicotinamide adenine dinucleotide seem to be reducing lifespan in high concentrations, this should be discussed
- It is stated that melatonin leads to reduced activity in the isolated individuals, similar trend could be seen (though not statistically important) in grouped individuals. One of the effects of the melatonin is to calm the body and regulate sleep cycle. Can it be, that melatonin is leading to less activity and thus reduces ROS, as the individuals are not moving so much? So in a way, we have “drugged” ants rather than just antioxidant treatment for social isolation?

- How were the social groups treatment maintained? Were dead individuals replaced to maintain the same number of ants? Or did the groups get smaller in time?

**< Responses to Reviewers >**

**Response to Reviewer 1**

Reviewer #1:

*Social animals are known to be stressed when kept in isolation. This experimental study*
*analyses the effect of social isolation on workers of the ant *Camponotus fellah*, showing*
*an increase in reactive oxygen species in isolated workers. The production of ROS is*
*reduced when workers are treated with melatonin, experimentally rescuing workers from*
*oxidative stress and stress related behaviours. This rescue has been shown before in the*
*solitary *Drosophila*, indicating that the link between ROS production, stress and*
*melatonin is not linked to sociality. The strength of the study is demonstrating*
*experimentally that short isolation (1 day) leads to ROS production in a social insect,*
*which in turn shortens the lifespan of ant workers. On the other hand, that social animals*
*including ants are stressed by isolation is not novel and also the behavioural analyses*
*only investigating the time close to the wall / in the nest, is not very revealing and detailed.*
*While the results are well written, the discussion could clearly be improved by a more*
*critical, multifaceted reflection of the findings.*

*Introduction*

*In general the introduction starts with a focus on human studies, and then introduces*
*social insects, as useful models to study the mechanistic effects of social isolation. This*
*framing suggests that the mechanisms underlying the negative effects of social isolation*
*are the same in humans, social mammals and insects. However, this might not be the case*
*and at least a critical reflection or citations of studies showing mechanistic similarities*
*would improve the introduction.*

*Line 44 References strongly focusses on human studies, even though it is a quite general*
*statement*

*We agree with the reviewer that there is little evidence that the mechanistic effects of*
*social isolation are the same in social insects, vertebrates and humans. However, it is*
*striking that social isolation induces similar consequences such as reduced lifespan,*
*impaired immunity, sleep disruption and metabolic dysfunction in these phylogenetically*
*distant organisms. We have modified the introduction to make this clear (lines 36-38).*

*Results*

*Line 76-78 The behavioural observations are not very detailed indeed the authors rather*
*report the location (next to the wall, in the nest) instead of the behaviours. The locational*

*shift can easily explained by isolated workers having to leave the nest for food or in*
*search for nestmates. Whether this “behaviour” is really a stress behaviour remain*
*unclear.*

We agree with the reviewer that we did not provide detailed behavioral data. To address
this, we now provide more information on the spatial location of the ants, dividing the
rearing boxes into 5 regions: nest, arena, near the walls, food area and water area, as
shown in the new Fig. S1a. Grouped ants spent most of their time in the nest, while
isolated ants were much more frequently near the walls or in the arena (see new Fig. 1a).
Isolated ants also spent more time in the water region, though the proportion of time spent
in the water region was low in both conditions. There was no significant difference in the
proportion of time workers spent in the food area, showing that the different space use of
isolated and grouped ants does not reflect a stronger tendency of isolated workers to leave
the nest in search of food.

We additionally conducted new analyses to quantify the activity level of isolated and
grouped workers. We determined the average speed and total distance covered per
individual (new data added to Fig. 1c). These analyses show that isolated ants moved
faster and traveled greater distances than grouped ants.

We also performed an additional experiment where we investigated whether ROS
production was correlated with the distance covered, the speed and/or space used, in
response to reviewer 3’s comments (see lines 279-291 on page 9 in this letter). We found
that there was no significant association between ROS production and distance covered
or speed. However, for isolated ants there was a positive correlation with the ratio of time
spent near the wall to time spent in the nest (a marker of stress) in Fig. 3c (first panel).
These new data are important because they demonstrate that increased ROS production
is not simply the result of increased activity. We have added these new data and
discussion of their significance (lines 180-191 and 290-293).

Overall, these new data show that social isolation not only affects the tendency to leave
the nest but also induces various behavioral changes including hyperactivity. We agree
with the reviewer that we should have been more careful about the cause of the behavioral
changes and now use the terms “hyperactivity” and “social isolation-induced behavior”
instead of “stress-related behavior” throughout the manuscript.

*Line 77 How long where ants isolated? Only 24h? Why this rather short social isolation*
*Time?*

We chose to isolate ants for 24h because social isolation leads to behavioral change within
this timeframe (Figs. 2b-2d in Koto et al., 2015) and because mortality starts to
significantly increase after 24h of isolation (Figs. 2a in Koto et al., 2015). We have
clarified the procedure on lines 74-77 and 609-614.

*Line 80 If you only record to location (next to wall, in nest) is it not unavoidable to find*
*a negative correlation?*

In our original analyses we had defined areas other than the nest and near the walls, so
the correlation was not unavoidable. But to address the reviewer's concern we have now
conducted analyses for each of the five areas: nest, arena, near the walls, food area, and
water area. These analyses show that isolated ants spend more time in the arena, near the
walls and in the water area but less time in the nest than grouped ants.

*Methods*

*Line 369 Why the different ages for the ants in the RNA seq and experimental studies?*
*Could that effect the interpretation of the results*

We used 4-month-old ants for the RNA-seq analyses because older individuals contain
more formic acid which hinders measurement of RNA concentration (see Appendix 1
below). Importantly, however, we examined the expression of four candidate genes with
qRTPCR in >7-month-old workers (Figs. 2b, d, f, and h) and found that gene expression
patterns were similar to 4-month-old workers, a point that was not well explained in the
original manuscript. Also, we previously found that isolation reduced lifespan
independently of age (Koto et al., 2015). We have clarified these two points in the revised
manuscript (lines 137-141, 423-428 and lines 527-529).

*Line 377 Specify the differences between ROS quantification in Fig 3a or Fig 4b*

The two figures have different aims. In Fig. 3a we compared ROS production between
 isolated and grouped individuals. By contrast, in the new Fig. 4c (Fig. 4b in the original
 manuscript), we compared ROS production between isolated individuals that had been
 given either water or water + melatonin. There were a few differences in the experimental
 design that are now explained in the method section (line 547-550).

*Line 418 Which amino acid sequences were analysed? What was the reason for this*
 *phylogenetic analyses? Please clarify.*

We performed the phylogenetic analysis to resolve the identity of the differentially
 expressed CYP genes. In insects, all CYPs are classified into one of the four clans: CYP2,
 CYP3, CYP4, and the mitochondrial CYP clan (Feyereisen, 2011). The differentially
 expressed CYP genes belonged to CYP clan 3 so we performed phylogenetic analysis of
 this clan specifically. The amino acid sequences were obtained from annotated transcript
 sequences with the reference genome of *Camponotus fellah*. The amino acid sequences
 of CYP genes in Hymenoptera were obtained from the Nelson cytochrome P450
 webpage⁶⁶, and the amino acid sequences in other species were obtained from NCBI using
 a blast search with the DEGs as queries. We have clarified this in the methods section
 (lines 479-484).

*Line 533 For how long where the ants isolated? It remains unclear, in which experiment,*
 *which isolation regime was done.*

In all experiments we kept ants in group or isolation for 24hr, during which we used the
behavioral tracking system to quantify their behavior. We have clarified this in the
method section (lines 609-610).

**Response to Reviewer 2**

Reviewer #2:

*Koto and colleagues describe behavioral, genetic, cellular, and molecular effects of*
*social isolation on the carpenter ant, Camponotus fellah. This work builds upon previous*
*studies that have demonstrated negative effects of social isolation in both social and non-*
*social taxa, including decreases in lifespan. However, uncovering the molecular and*
*cellular underpinnings of aging phenotypes has been challenged by variable results*
*between studies – which in many cases may reflect true differences between experimental*
*conditions and taxa. Relatively few studies go beyond correlation to demonstrate causal*
*effects of the genes or molecules identified. By following up on changes in gene expression*
*in socially isolated workers, Koto et al. identify tissue-specific ROS damage in socially-*
*isolated ants that they rescue pharmacologically. The authors confirm an earlier study*
*that showed socially isolated workers spend less time near the nest and more time near*
*the walls of an arena – an atypical behavior likely indicating stress. Their analysis of*
*differential gene expression identifies an enrichment in genes related to oxidoreductase*
*activity, consistent with ideas that reactive oxygen species (ROS) are causally involved*
*in many senescence-associated phenotypes. Using RNA-seq and qRT-PCR, they find*
*significant differences in several ROS-associated genes, notably DUOX, an NADPH-*
*oxidase, and three genes involved in detoxification. The authors next confirmed higher*
*levels of ROS in fatbodies and oenocytes, but no significant difference in the heads and*
*digestive system of the ants. This tissue-specificity is consistent with other studies, several*
*of which show lack of oxidative damage in the nervous system with aging. Koto and*
*colleagues conclude the paper with the most compelling result – they significantly*
*increase lifespan in socially isolated ants by feeding them antioxidants. This finding*
*demonstrates quite convincingly that oxidative stress is causally involved in the shortened*
*lifespans of socially-isolated ants. This work makes an important contribution to our*
*understanding of aging in both eusocial and non-eusocial taxa and provides specific*
*molecular pathways that should be examined in other systems. The findings are novel and*
*of broad interest to researchers focused on the physiology of aging, social insects, and*
*mechanisms involved in senescence.*

We are pleased that the reviewer found our study interesting.

*However, before publication, several aspects of the manuscript require clarification or*
*additional details. I have one area of significant methodological concern, regarding the*
*quantification of fluorescence. The intensity of fluorescence obtained from a confocal*
*microscope can be strongly influenced by imaging parameters, such as gain and offset.*
*Generally, quantifying intensity can be problematic, especially if all images are not*
*acquired under the same conditions. The methods section neither describes the imaging*
*parameters used nor whether they were consistent across treatments. While the data from*
*the ROS quantification (Fig 3A) are convincing, I am not sure how to interpret the data*
*presented in Fig 3B. If the images were not acquired under the same conditions, they*
*could potentially be normalized to the DAPI intensity to allow comparisons between*
*treatments. The authors should provide additional details and perhaps analysis to*
*demonstrate that quantitative comparisons can be made between different images.*

We agree that we did not provide sufficient information on how we quantified the
intensity of fluorescence. To allow reliable comparisons between images we always used
the same conditions (exposure time, gain, laser power, etc). We have now added this
information in the method section (lines 578-580).

*My other concerns and comments are minor, and I address these aspects line by line*
*below. If all concerns are addressed, I strongly recommend publication for this*
*interesting and important work.*

*Line 52: The authors should broaden the context of their work to include more references*
*to aging in social insects outside of Camponotus. Notably studies on other ant species*
*and bees would be useful to reference here.*

We agree with the reviewer's suggestion. We have now added a new section to discuss
results of studies in other social insects (lines 43-50).

*Line 184: It is possible that the effects of antioxidants are not specific to social isolation,*
*but could act to increase lifespan under conditions of high ROS/low antioxidant activity,*
*which in this case was induced by social isolation.*

This is a good point. To avoid confusion, we deleted the word "specifically" and clarified
the last sentence in this paragraph (lines 207-209).

*Line 218: Change hydrocarbon to hydrocarbons.*

Changed as suggested by the reviewer (line 243).

*Line 239: Is this supposed to be a new paragraph or just new sentence?*

It was supposed to be a new paragraph. We have now corrected this (lines 265-274).

*Line 254: As mentioned for line 184 above, it is possible that rescue effects of antioxidants*
*in socially-isolated workers does not indicate that they only act during social isolation,*
*but rather, exert an effect on lifespan only when under high ROS/low antioxidant*
*conditions. Other stressors could similarly shorten lifespan but then be rescued by such*
*a treatment, but that of course awaits further study.*

We agree with the reviewer that the antioxidants could have a rescue effect under
circumstances other than social isolation. To clarify this, we have now modified the
sentence to make it clear that our claim relates to our experiments (lines 285-287).

*Line 396: Why were 4- and 5-month-old ants used for the RNA-seq analysis while 7-*
*month olds used for all other aspects of the study? Given the importance of age on the*
*phenotypes measured it would be useful to know why this was done and if there is any*
*significant difference in gene expression between these two age groups.*

This is a very important point which was also made by the two other reviewers. For RNA-
seq analyses we used 4-month-old individuals because older individuals contain more
formic acid, which hinders measurement of RNA concentration (see Appendix 1 on page
4 in this letter). Importantly, however, we examined the expression of four candidate
genes with qRTPCR in 7-month-old workers (Figs. 2b, d, f, and h) and found that gene
expression patterns were similar for these individuals and 4-month-old workers, a point
that was not well explained in the original manuscript. Also, we previously found that
isolation reduced lifespan independently of age (Koto et al., 2015). We have clarified both
points in the manuscript (lines 137-141, 423-428 and lines 527-529).

*Line 496: Even though the method is described in another paper, it would be helpful to*

*briefly state the incubation times and temperatures for the primary and secondary*
*antibodies.*

We agree and have added this information (temperature and duration for staining) on lines
567-571.

*Line 534: Please provide a brief description of the behavioral tracking assay, space*
*allowing. Although the details can be found elsewhere, it would be very helpful to have a*
*1-2 sentence description of the tracking and analysis. It would be interesting to know if*
*other behaviors than time in nest and time at the wall were tracked. If so, did these show*
*a significant effect of treatment? Such data could appear in the supplement.*

This is a good point. We now provide information on the behavioral tracking assay (lines
74-79, 608-624) and conducted some new analyses on the duration of time spent in
different areas of the nest/foraging arena (new Fig. 1a), as well as the speed and distance
covered by grouped and isolated ants (new Fig. 1c).

*Figure 4: The yellow survival curves on the white background for the middle panel (4a)*
*are hard to see. I would recommend selecting a different color.*

We agree and now use green instead of yellow in Fig 4a.

**Response to Reviewer 3**

Reviewer #3:

*The manuscript is based on the experiment carried out with an ant *Camponotus fellah**
*and studies the effect of social isolation on the oxidative stress in the worker ants. The*
*authors use different molecular, histological methods as well as bioassays to examine the*
*role of the social isolation and the effect of ROS on the longevity in the ant workers. They*
*claim that the release of free radicals is the cause of reduced life span in the isolation*
*and feeding antioxidants will decrease the ROS-induced mortality in the social isolation.*
*While the topic is interesting in the ant world, all in all I feel that authors are*
*unfortunately oversimplifying the results and the data collection for the claims is not done*
*properly. Please see my comments below.*

*The way the experiments are carried out raises several concerns:*

*- Much of the conclusions are based on the RNA-sequencing data, which is based on the*

*material originating from ants, which have been in isolation for 24h. At the same it time,*
*it is stated that social isolation leads to higher activity levels – workers are moving*
*around more. One would argue that the higher levels of ROS in the organism are linked*
*to the physical activity (which is known to generate ROS) rather than to the social*
*isolation. Even if the higher activity levels are due to the ant worker being in isolation,*
*the cause increased ROS is still physical activity.*

We thank the reviewer for this suggestion. It is true that the higher level of ROS in isolated
ants could be due to their increased activity. To address this issue, we performed
additional experiments to examine the correlation between activity level and ROS in
grouped and isolated ants (new Fig. 3c). We performed behavioral tracking of grouped
and isolated workers and then quantified ROS markers (CellROX DeepRed) in the fat
body and oenocytes of each ant. These analyses confirmed a positive correlation between
ROS intensity and the ratio of time spent near the wall to time spent in the nest (1st panel
in Fig. 3c). Importantly, this correlation was specifically observed in isolated ants but not
in grouped ants. Moreover, these new analyses revealed no significant correlation
between ROS production and activity level (speed nor distance covered) in either grouped
or isolated ants (2nd and 3rd panel in Fig. 3c). We believe that these new data should fully
address the reviewer's concern. These results are now shown in Fig. 3c and described on
lines 180-191 and lines 556-558.

*- As the RNA-sequencing was carried out in 4-5 month old ants, and the rest of the studies*
*in much older individuals (more than 7) it really is impossible to extrapolate any results*
*over these age classes. Even, if the workers are relatively long-lived (to which I did not*
*find any reference to in the text), one can expect certain age-related physiological*
*changes taking place in the organism. It is known, that one of the ageing related effects*
*is the accumulation of ROS-related damages. Hence, it is of course not a surprise, that*
*feeding antioxidants can alleviate this effect.*

We agree that it is important to address the issue of age differences of individuals used in
the different experiments (a point also made by the two other reviewers). For RNA-seq
analysis we used 4-month-old individuals because older individuals contain more formic
acid which hinders measurement of RNA concentration (see Appendix 1 in response to
reviewer 1, on page 4 in this letter). Importantly, however, we examined the expression
of four candidate genes with qRT-PCR in 7-month-old workers (Figs. 2b, d, f, and h) and
found that gene expression patterns were similar to 4-month-old workers, a point that was

not well explained in the original manuscript. Also, we previously found that isolation
reduced lifespan independently of age (Koto et al., 2015). We have clarified both points
in the manuscript (lines 137-141, 423-428 and lines 527-529).

We agree with the reviewer that oxidative stress is a physiological change associated with
aging in various animals. However, the aim of our study was not to compare young and
old individuals. We studied the effect of social isolation, and our results show that
antioxidants can rescue the oxidative stress and the higher mortality associated with social
isolation among age-matched workers. Importantly, there was no statistical difference in
longevity between control and antioxidant-treated ants in the grouped condition (Fig. 4a).
If anything, there was a tendency (not significant) for treatments with the highest dosage
of melatonin (1 μ g/ml) and both dosages of NAD (0.13 μ g/ml and 0.43 μ g/ml) to reduce
the lifespan of grouped ants. We have added this information to the manuscript (lines
205-209 and 284-285).

*- Alternative explanations for the life-span differences should be considered. It is well*
*possible, that trophallaxis, which was not available for isolated ants can transfer certain*
*molecules which are beneficial, the lack of access to this can be detrimental. And, while*
*it is of course linked to isolation it is not a direct cause.*

We agree with the reviewer. We added this possibility in the discussion (lines 272-274).

*- It is a bit hard to follow the timeline and ages of the ants for different assays. The*
*manuscript would benefit from the addition of a figure which highlights how the samples*
*were taken and when the samples were taken for different analysis.*

We agree that the timeline and ages of the individuals used were not explained clearly.
We have added this information on the top of Fig. 2 and lines 137-141 and 423-428. We
also added information on sample sizes for all experiments/treatments in either the graphs
or figure legends.

*- Both melatonin and nicotinamide adenine dinucleotide seem to be reducing lifespan in*
*high concentrations, this should be discussed*

This is a good point. Although the treatment with higher concentration of the two
antioxidants (Mel 10 μ g/ml or NAD 1.3 μ g/ml) did not significantly affect the survival of

isolated ants, grouped ants treated with Mel 10 μ g/ml and NAD 1.3 μ g/ml tended to be
shorter-lived. We have now added these observations to the manuscript (lines 205-209
and 284-285).

*- It is stated that melatonin leads to reduced activity in the isolated individuals, similar*
*trend could be seen (though not statistically important) in grouped individuals. One of*
*the effects of the melatonin is to calm the body and regulate sleep cycle. Can it be, that*
*melatonin is leading to less activity and thus reduces ROS, as the individuals are not*
*moving so much? So in a way, we have “drugged” ants rather than just antioxidant*
*treatment for social isolation?*

We thank the reviewer for pointing out this issue. To address this question, we have
performed new behavioral analyses to examine whether other activity parameters were
also affected by melatonin treatment. We calculated the total distance covered and the
speed of all individuals. These analyses showed no significant effect of melatonin on
these parameters for either grouped or isolated ants (Fig. 4d). We added these new data
to Fig. 4d and discuss them on lines 215-218. We also performed an additional experiment
where we investigated whether ROS production was correlated with the distance covered,
the speed and/or space used. We found that there was no significant association between
ROS production and distance covered or speed (2nd and 3rd panel in Fig. 3c). However,
for isolated ants there was a positive correlation with the ratio of time spent near the wall
to time spent in the nest (a marker of stress) in Fig. 3c (1st panel). These new data are
important because they demonstrate that increased ROS production is not simply the
result of increased activity. We have added these new data and discussion of their
significance (lines 180-191 and 290-293).

We now have also included a new section where we discuss the similarity of behavioral
changes induced by social isolation in ants and rodents (lines 293-299). In rodents, it is
standard practice to use an open field test which measures the time spent by mice in the
peripheral or central zones of a box and calculates the distance traveled by video
recording. Experiments show that increased anxiety results in decreased exploratory
behavior in the central zone and a preference to stay near the walls of box (Berry et al.,
2012, Ieraci et al., 2016), as well as a greater distance traveled in the peripheral zone and
a shorter distance in the central zone (Ieraci et al., 2016). Although the experimental
designs were not identical, it is interesting to note that social isolation in both ants and
mice caused individuals to stay closer to the periphery of their boxes.

- *How were the social groups treatment maintained? Were dead individuals replaced to*
*maintain the same number of ants? Or did the groups get smaller in time?*

We never replaced any individuals. Given that we previously found (Koto et al., 2015)
that mortality starts to increase after 24 hours of isolation we only conducted 24-hour
experiments to avoid this problem. The mortality under grouped conditions was always
very low (e.g., one worker for the RNAseq experiments).

Reviewers' Comments:

Reviewer #2:

Remarks to the Author:

Koto and colleagues have made significant changes to the manuscript, including additional data, new analyses, and more detailed explanations for experiments. The additional behavioral analyses strengthen the authors' argument that social isolation leads to behavioral changes in space use and leads to hyperactivity. My primary concerns from the original submission were the quantification of fluorescence and age differences between ants used in most experiments and the RNAseq data. I requested additional clarity and methodological detail throughout the paper. For the most part, the authors have satisfied my concerns. However, before I can recommend publication, some additional revisions are necessary.

My primary concern still rests on the fact that the authors conducted all experiments on 7-month old ants, but the RNA-seq data were collected on 4-month old ants. This methodological difference is particularly evident in a paper on ROS, when age is a common factor that influences antioxidant and ROS levels, and the authors even conclude their paper discussing the role of antioxidants in lifespan extension in isolated ants. Although the authors have explained clearly why it was not feasible to run the RNA-seq experiments on 7-month old ants due to high concentrations of formic acid, I feel that the manuscript needs to clarify differences that were observed in the RNAseq and RTqPCR data sets (Fig 2). Although overall patterns are similar, differences in CYP336A26 were notable. While large differences were seen in the data on 4 month old ants, there was only a slight difference in the abdomen in 7-month old ants and no differences in any other tissue. This should be discussed in more detail in the text and possible explanations offered (ex. Lines 137-145).

I would like to see a few additional concerns addressed as detailed below:

Lines 78-82 (Fig1a): While I understand that the authors presented ratios of time spent in various regions to deal with the fact that most ants were in the nest, the actual times should be reported somewhere (such as a supplement). I would like to see an analysis of total time (perhaps excluding the in-nest time) to see if it shows the same pattern as the ratio data.

Line 139: While explained elsewhere, you need to explain to the reader here why you collected data on different ages. Otherwise it jumps out as an odd experimental choice.

Lines 134-136, 139-141: Although there are many similarities in the RNAseq and RTqPCR datasets, in both of these sections of text, it seems as though real differences are not being noted. The pattern is NOT the same in all cases, as I would not say that a significant difference and a nonsignificant difference are "in the same direction". These sentences should be revised accordingly to reflect the differences that do exist in the data.

Lines 183-187: While the correlations are helpful, the correlation coefficients are very small, indicating that they do not explain much of the behavioral data. This should be discussed here. Also, please report the R^2 values in the text, along with the F and p-values.

Line 273: add "perform" in front of trophallaxis

Reviewer #3:

None

Reviewer #4:

Remarks to the Author:

This is the first time I read the manuscript entitled "Social isolation shortens lifespan through oxidative stress in ants". I also read the former reviews and the authors' replies. I think that the authors answered well most comments, and the manuscript has been greatly improved. I also believe the experiments are well conducted. In my opinion, the results are sufficiently interesting and novel to be published more or less as presented now and I recommend accepting the manuscript following a minor revision. I suggest improving two points: (a) I agree with the reviewers that the literature on isolation in eusocial insects and ants in particular is not covered very well. I brought several suggestions on how to improve it. This is a minor issue, though. (b) I think that the way the authors present their findings in the summary and end of the introduction should be improved. I think that the important points are somewhat neglected. See my comment regarding L 60-69 below. But all in all, this is a very good manuscript, which I read with interest, and I would be happy to see it published.

L 38-39: I agree that relatively little is known about the mechanisms behind the negative effects of social isolation. However, recent papers have begun to test them, and it would be good to acknowledge them here and describe them in a sentence or two. There are already several papers doing so.

L 43: Add "mostly" before "(ants, some bees..." because there are other eusocial insects, which you did not mention.

L 57-58: This is not 100% accurate. Regarding *T. nylanderi*, there is a suggested mechanism in Scharf et al. (2021 *Molecular Ecology*), which is cited later. The imbalance in energy intake suggested for *C. fellah* sounds to me also as a suitable and logical mechanism. Regarding digestion disruptions, perhaps see also Howard and Tschinkel (1981 *Journal of Insect Physiology*) or Gosswald and Kloft (1960 *Biocontrol*). There are also papers reporting on flawed immune system function following isolation, which is also a mechanism explaining some of the negative consequences of social isolation (e.g., Traniello et al. 2002 *PNAS*, but there are probably more papers). Finally, some studies examined how neurotransmitters in the brain change with isolation. Changes in the levels of dopamine, serotonin, or octopamine were documented and all can serve as a mechanism for behavioral changes under isolation (e.g., Wada-Katsumata et al. 2011 *Journal of Experimental Biology*; Tsvetkov et al. 2019 *Journal of Experimental Biology*).

L 60-69: The novelty of the current study is not 100% clear here. This is unfortunate because the results are very interesting. First, I think that the result in lines 135-136 (not only comparing isolated vs. grouped workers but you manage to show a difference in gene expression within the group of isolated workers and link it to gene expression) can be used here to emphasize the novelty of this paper. Second, you used different body parts for gene expression, which is also new (L 142-145). Third, on top of gene expression, you used a biochemical method to quantify oxidative stress. This is another strong point, which is not clear enough in the summary or the end of the introduction. Fourth, the usage of melatonin as an antioxidant should be mentioned here too. I would more clearly emphasize these points. The study is more novel than how it is perceived by reading the abstract or the introduction's end.

L 77: Does a group of 10 represent well the situation in the colony? I guess there is no queen or brood, so does it represent well the normal situation?

L 86-88: As one of your results is increased activity, I would recommend referring to previous studies reporting on changes in activity with isolation. See, for instance, Ahronberg & Scharf (2021 *Behavioural Processes*, Mildner & Roces 2017 *PLoS One*, or McCarthy et al. 2015 *Journal of Insect Behavior*).

L 199: This is the first time melatonin is mentioned. I would mention it before, in the introduction, when describing what has been done.

L 434-435: Two-Four colonies strike me as a low number. This makes all groups dependent on each other. Colony ID should be taken as a random factor in all analyses. It is not 100% clear to me if this is the case. If yes, this is good, please write it explicitly. If not, please correct it.

**< Responses to Reviewers >**

**Response to Reviewer 2**

**Reviewer #2 (Remarks to the Author):**

*Koto and colleagues have made significant changes to the manuscript, including additional data,*
*new analyses, and more detailed explanations for experiments. The additional behavioral*
*analyses strengthen the authors' argument that social isolation leads to behavioral changes in*
*space use and leads to hyperactivity. My primary concerns from the original submission were the*
*quantification of fluorescence and age differences between ants used in most experiments and the*
*RNAseq data. I requested additional clarity and methodological detail throughout the paper. For*
*the most part, the authors have satisfied my concerns. However, before I can recommend*
*publication, some additional revisions are necessary.*

*My primary concern still rests on the fact that the authors conducted all experiments on 7-month*
*old ants, but the RNA-seq data were collected on 4-month old ants. This methodological*
*difference is particularly evident in a paper on ROS, when age is a common factor that influences*
*antioxidant and ROS levels, and the authors even conclude their paper discussing the role of*
*antioxidants in lifespan extension in isolated ants. Although the authors have explained clearly*
*why it was not feasible to run the RNA-seq experiments on 7-month old ants due to high*
*concentrations of formic acid, I feel that the manuscript needs to clarify differences that were*
*observed in the RNAseq and RTqPCR data sets (Fig 2). Although overall patterns are similar,*
*differences in CYP336A26 were notable. While large differences were seen in the data on 4 month*
*old ants, there was only a slight difference in the abdomen in 7-month old ants and no differences*
*in any other tissue. This should be discussed in more detail in the text and possible explanations*
*offered (ex. Lines 137-145).*

**We thank the reviewer for his/her positive review and useful suggestions. We agree that it is**
**important to clarify the differences that were observed between the RNA-seq and qRTPCR data**
**sets for CYP336A26. We have now added a paragraph to discuss the difference (lines 184-193).**

*I would like to see a few additional concerns addressed as detailed below:*

*Lines 78-82 (Fig1a): While I understand that the authors presented ratios of time spent in various*
*regions to deal with the fact that most ants were in the nest, the actual times should be reported*
*somewhere (such as a supplement). I would like to see an analysis of total time (perhaps excluding*
*the in-nest time) to see if it shows the same pattern as the ratio data.*

We followed the reviewer's suggestion and now provide the data on the time (min/day) spent in
the nest, near the wall, in the arena, in the food area, and in the water area of grouped and isolated
ants in a new panel (b) in Supplementary figure (Fig. S1). The pattern is similar to the ratio data
presented in Fig. 1a. We have now added this information in line 103 and explained the new data
of this figure in lines 103-106.

*Line 139: While explained elsewhere, you need to explain to the reader here why you collected*
*data on different ages. Otherwise it jumps out as an odd experimental choice.*

We agree with the reviewer that an explanation is needed for why we used 4-month-old and 7-
45 month-old workers. We previously showed that social isolation has similar negative effects on
lifespan regardless of worker age (Koto et al., 2015). However, the average lifespan of isolated
4-month-old workers is about 50 days while it is about 20 days for >7-month-old workers (Koto
et al., 2015). This is why we decided to use >7-month-old workers in the experiment. The only
problem with the older workers is that they contain high levels of formic acid which decreases
the amount of purified RNA available when conducting the extractions for the RNA-seq analyses
(but this is not a problem for qRTPCR and ROS quantifications). We have now provided this
information in lines 98-102 and 140-143.

*Lines 134-136, 139-141: Although there are many similarities in the RNAseq and RTqPCR*
*datasets, in both of these sections of text, it seems as though real differences are not being noted.*
*The pattern is NOT the same in all cases, as I would not say that a significant difference and a*
*nonsignificant difference are "in the same direction". These sentences should be revised*
*accordingly to reflect the differences that do exist in the data.*

We agree with the reviewer's comment. We have rewritten this section and added a new paragraph
(lines 184-193) to provide possible explanations for the difference in the RNA-seq and qRTPCR
data, especially for CYP336A26.

*Lines 183-187: While the correlations are helpful, the correlation coefficients are very small,*
*indicating that they do not explain much of the behavioral data. This should be discussed here.*
*Also, please report the R² values in the text, along with the F and p-values.*

We agree and now provide the R^2 -values along the F - and p -values in the manuscript (lines 222-
226). We have also made clear that the correlation between CellROX intensity and wall:nest ratio
is relatively small, suggesting that other intrinsic factors must be involved (lines 230-231).

*Line 273: add “perform” in front of trophallaxis*

Done as suggested in line 313.

**Response to Reviewer 4**

**Reviewer #4 (Remarks to the Author):**

*This is the first time I read the manuscript entitled "Social isolation shortens lifespan through*
*oxidative stress in ants". I also read the former reviews and the authors' replies. I think that the*
*authors answered well most comments, and the manuscript has been greatly improved. I also*
*believe the experiments are well conducted. In my opinion, the results are sufficiently interesting*
*and novel to be published more or less as presented now and I recommend accepting the*
*manuscript following a minor revision. I suggest improving two points: (a) I agree with the*
*reviewers that the literature on isolation in eusocial insects and ants in particular is not covered*
*very well. I brought several suggestions on how to improve it. This is a minor issue, though. (b) I*
*think that the way the authors present their findings in the summary and end of the introduction*
*should be improved. I think that the important points are somewhat neglected. See my comment*
*regarding L 60-69 below. But all in all, this is a very good manuscript, which I read with interest,*
*and I would be happy to see it published.*

We are pleased that the reviewer found our study interesting and novel. We also found their
comments very useful and have implemented all the suggestions in the revised manuscript.

*L 38-39: I agree that relatively little is known about the mechanisms behind the negative effects*
*of social isolation. However, recent papers have begun to test them, and it would be good to*
*acknowledge them here and describe them in a sentence or two. There are already several papers*
*doing so.*

We agree with the reviewer that several studies have started to analyze the mechanisms. We have
added this information in lines 40-42 and lines 70-75.

*L 43: Add "mostly" before "(ants, some bees..." because there are other eusocial insects, which*
*you did not mention.*

We agree and added “mostly”, as suggested (line 46).

*L 57-58: This is not 100% accurate. Regarding T. nylanderi, there is a suggested mechanism in*
*Scharf et al. (2021 Molecular Ecology), which is cited later. The imbalance in energy intake*
*suggested for C. fellah sounds to me also as a suitable and logical mechanism. Regarding*
*digestion disruptions, perhaps see also Howard and Tschinkel (1981 Journal of Insect*
*Physiology) or Gosswald and Kloft (1960 Biocontrol). There are also papers reporting on flawed*
*immune system function following isolation, which is also a mechanism explaining some of the*
*negative consequences of social isolation (e.g., Traniello et al. 2002 PNAS, but there are probably*
*more papers). Finally, some studies examined how neurotransmitters in the brain change with*
*isolation. Changes in the levels of dopamine, serotonin, or octopamine were documented and all*
*can serve as a mechanism for behavioral changes under isolation (e.g., Wada-Katsumata et al.*
*2011 Journal of Experimental Biology; Tsvetkov et al. 2019 Journal of Experimental Biology).*

We thank the reviewer for pointing out that this section was not clear. We have made significant
changes in the introduction to address this. We now mention that many social processes are also
involved in altering the lifespan of socially-isolated individuals (lines 62-68). We also now make
clear that social isolation affects the expression of genes related to immune functions, stress
response and levels of biogenic amines (new paragraph in lines 70-75).

*L 60-69: The novelty of the current study is not 100% clear here. This is unfortunate because the*
*results are very interesting. First, I think that the result in lines 135-136 (not only comparing*
*isolated vs. grouped workers but you manage to show a difference in gene expression within the*
*group of isolated workers and link it to gene expression) can be used here to emphasize the novelty*
*of this paper. Second, you used different body parts for gene expression, which is also new (L*
*142-145). Third, on top of gene expression, you used a biochemical method to quantify oxidative*
*stress. This is another strong point, which is not clear enough in the summary or the end of the*

*introduction. Fourth, the usage of melatonin as an antioxidant should be mentioned here too. I*
*would more clearly emphasize these points. The study is more novel than how it is perceived by*
*reading the abstract or the introduction's end.*

*We thank the reviewer for pointing out that the novelty of our study was not well explained in the*
*summary and introduction. We extended the last paragraph of the introduction according to the*
*reviewer's comments (lines 81-82 and 84-86).*

*L 77: Does a group of 10 represent well the situation in the colony? I guess there is no queen or*
*brood, so does it represent well the normal situation?*

*The reviewer is correct that the lifespan of workers does not only depend on the number of*
*workers but also on other factors. For example, we (and others) had found that the presence of*
*brood increases lifespan. This is now clearly stated in lines 57, and 66-68.*

*L 86-88: As one of your results is increased activity, I would recommend referring to previous*
*studies reporting on changes in activity with isolation. See, for instance, Ahronberg & Scharf*
*(2021 Behavioural Processes, Mildner & Roces 2017 PLoS One, or McCarthy et al. 2015 Journal*
*of Insect Behavior).*

*We agree and have added a sentence (lines 107-108) to mention that changes in activity levels of*
*isolated individuals have also been reported in other social and non-social insects.*

*L 199: This is the first time melatonin is mentioned. I would mention it before, in the introduction,*
*when describing what has been done.*

*We agree and now mention that we used melatonin as one of the antioxidant compounds in the*
*introduction (line 88).*

*L 434-435: Two-Four colonies strike me as a low number. This makes all groups dependent on*
*each other. Colony ID should be taken as a random factor in all analyses. It is not 100% clear to*
*me if this is the case. If yes, this is good, please write it explicitly. If not, please correct it.*

- This is a very important point. We always considered colony ID as a random factor in the analyses.
- We have clarified this in lines 490-492.

Reviewers' Comments:

Reviewer #2:

Remarks to the Author:

I thank the authors for their revisions, and feel that the changes to text and additional data have strengthened the manuscript. These changes have satisfied all my concerns, and I find the manuscript ready for publication.

Reviewer #4:

Remarks to the Author:

This is the second time I read the manuscript. All my comments were answered well. I have no further comments and congratulate the authors for their interesting work. I recommend on accepting the manuscript for publication.